# Dopamine role in learning and action inference

**Rafal Bogacz\***

MRC Brain Networks Dynamics Unit, University of Oxford, Oxford, United Kingdom

**Abstract** This paper describes a framework for modelling dopamine function in the mammalian brain. It proposes that both learning and action planning involve processes minimizing prediction errors encoded by dopaminergic neurons. In this framework, dopaminergic neurons projecting to different parts of the striatum encode errors in predictions made by the corresponding systems within the basal ganglia. The dopaminergic neurons encode differences between rewards and expectations in the goal-directed system, and differences between the chosen and habitual actions in the habit system. These prediction errors trigger learning about rewards and habit formation, respectively. Additionally, dopaminergic neurons in the goal-directed system play a key role in action planning: They compute the difference between a desired reward and the reward expected from the current motor plan, and they facilitate action planning until this difference diminishes. Presented models account for dopaminergic responses during movements, effects of dopamine depletion on behaviour, and make several experimental predictions.

## Introduction

Neurons releasing dopamine send widespread projections to many brain regions, including basal ganglia and cortex (*Björklund and Dunnett, 2007*), and substantially modulate information processing in the target areas. Dopaminergic neurons in the ventral tegmental area respond to unexpected rewards (*Schultz et al., 1997*), and hence it has been proposed that they encode reward prediction error, defined as the difference between obtained and expected reward (*Houk et al., 1995*; *Montague et al., 1996*). According to the classical reinforcement learning theory, this prediction error triggers update of the estimates of expected rewards encoded in striatum. Indeed, it has been observed that dopaminergic activity modulates synaptic plasticity in the striatum in a way predicted by the theory (*Reynolds et al., 2001*; *Shen et al., 2008*). This classical reinforcement learning theory of dopamine has been one of the greatest successes of computational neuroscience, as the predicted patterns of dopaminergic activity have been seen in diverse studies in multiple species (*Eshel et al., 2016*; *Tobler et al., 2005*; *Zaghloul et al., 2009*).

However, this classical theory does not account for the important role of dopamine in action planning. This role is evident from the difficulties in initiation of voluntary movements seen after the death of dopaminergic neurons in Parkinson's disease. This role is consistent with the diversity in the activity of dopaminergic neurons, with many of them responding to movements (*da Silva et al., 2018*; *Dodson et al., 2016*; *Howe and Dombeck, 2016*; *Jin and Costa, 2010*; *Lee et al., 2019*; *Schultz et al., 1983*; *Syed et al., 2016*). The function of dopamine in energizing movements is likely to come from the effects it has on the excitability or gain of the target neurons (*Lahiri and Bevan, 2020*; *Thurley et al., 2008*). Understanding the role of dopamine in action planning and movement initiation is important for refining treatments for Parkinson's disease, where the symptoms are caused by dopamine depletion.

A foundation for a framework accounting the role of dopamine in both learning and action planning may be provided by a theory called active inference (*Friston, 2010*). This theory relies on an assumption that the brain attempts to minimize prediction errors defined as the differences between

**\*For correspondence:**
rafal.bogacz@ndcn.ox.ac.uk

**Competing interests:** The author declares that no competing interests exist.

**eLife digest** In the brain, chemicals such as dopamine allow nerve cells to 'talk' to each other and to relay information from and to the environment. Dopamine, in particular, is released when pleasant surprises are experienced: this helps the organism to learn about the consequences of certain actions. If a new flavour of ice-cream tastes better than expected, for example, the release of dopamine tells the brain that this flavour is worth choosing again.

However, dopamine has an additional role in controlling movement. When the cells that produce dopamine die, for instance in Parkinson's disease, individuals may find it difficult to initiate deliberate movements. Here, Rafal Bogacz aimed to develop a comprehensive framework that could reconcile the two seemingly unrelated roles played by dopamine.

The new theory proposes that dopamine is released when an outcome differs from expectations, which helps the organism to adjust and minimise these differences. In the ice-cream example, the difference is between how good the treat is expected to taste, and how tasty it really is. By learning to select the same flavour repeatedly, the brain aligns expectation and the result of the choice. This ability would also apply when movements are planned. In this case, the brain compares the desired reward with the predicted results of the planned actions. For example, while planning to get a spoonful of ice-cream, the brain compares the pleasure expected from the movement that is currently planned, and the pleasure of eating a full spoon of the treat. If the two differ, for example because no movement has been planned yet, the brain releases dopamine to form a better version of the action plan. The theory was then tested using a computer simulation of nerve cells that release dopamine; this showed that the behaviour of the virtual cells closely matched that of their real-life counterparts.

This work offers a comprehensive description of the fundamental role of dopamine in the brain. The model now needs to be verified through experiments on living nerve cells; ultimately, it could help doctors and researchers to develop better treatments for conditions such as Parkinson's disease or ADHD, which are linked to a lack of dopamine.

observed stimuli and expectations. In active inference, these prediction errors can be minimized in two ways: through learning – by updating expectations to match stimuli, and through action – by changing the world to match the expectations. According to the active inference theory, prediction errors may need to be minimized by actions, because the brain maintains prior expectations that are necessary for survival and so cannot be overwritten by learning, e.g. an expectation that food reserves should be at a certain level. When such predictions are not satisfied, the brain plans actions to reduce the corresponding prediction errors, for example by finding food.

This paper suggests that a more complete description of dopamine function can be gained by integrating reinforcement learning with elements of three more recent theories. First, taking inspiration from active inference, we propose that prediction errors represented by dopaminergic neurons are minimized by both learning and action planning, which gives rise to the roles of dopamine in both these processes. Second, we incorporate a recent theory of habit formation, which suggests that the habit and goal-directed systems learn on the basis of distinct prediction errors (*Miller et al., 2019*), and we propose that these prediction errors are encoded by distinct populations of dopaminergic neurons, giving rise to the observed diversity of their responses. Third, we assume that the most appropriate actions are identified through Bayesian inference (*Solway and Botvinick, 2012*), and present a mathematical framework describing how this inference can be physically implemented in anatomically identified networks within the basal ganglia. Since the framework extends the description of dopamine function to action planning, we refer to it as the DopAct framework. The DopAct framework accounts for a wide range of experimental data including the diversity of dopaminergic responses, the difficulties in initiation of voluntary movements under dopamine depletion, and it makes several experimentally testable predictions.

## Results

To provide an intuition for the DopAct framework, we start with giving its overview. Next, we formalize the framework, and show examples of models developed within it for two tasks commonly used

in experimental studies of reinforcement learning and habit formation: selection of action intensity (such as frequency of lever pressing) and choice between two actions.

## Overview of the framework

This section first gives an overview of computations taking place during action planning in the DopAct framework, and then summarizes how these computations could be implemented in neural circuits including dopaminergic neurons.

The DopAct framework includes two components contributing to planning of behaviour. The first component is a valuation system, which finds the value $v$ of reward that the animal should aim at acquiring in a given situation. A situation of an animal can be described by two classes of factors: internal factors connected with level of reserves such as food, water, etc. to which we refer as 'reserves', and external factors related to the environment, such as stimuli or locations in space, to which we refer as a 'state' following reinforcement learning terminology. The value $v$ depends on both the amount of reward available in state $s$, and the current level of reserves. For example, if animal is not hungry, the desired value is equal to $v = 0$ even if food is available. The second component of the DopAct framework is an actor, which selects an action to obtain the desired reward. This paper focusses on describing computations in the actor. Thus, for simplicity, we assume that the valuation system is able to compute the value $v$, but this paper does not describe how that computation is performed. In simulations we mostly focus on a case of low reserves, and use a simple model similar to a critic in standard reinforcement learning, which just learns the average value $v(s)$ of resource in state $s$ (**Sutton and Barto, 1998**). Extending the description of the valuation system will be an important direction for future work and we come back to it in Discussion.

The goal of the actor is to select an action to obtain the reward set by the valuation system. This action is selected through inference in a probabilistic model, which describes relationships between states, actions and rewards, which we denote by $s$, $a$ and $R$. Following reinforcement learning convention, we use $R$ to denote the total reward defined in Equation 1.1 of **Figure 1A**, which includes the current reward $r$, and the future reward value $v$ computed by the valuation system. The DopAct framework assumes that two systems within the actor learn distinct relationships between the variables, shown in **Figure 1A**. The first system, shown in orange, learns how the reward depends on the action selected in a given state, and we refer to it as 'goal-directed', because it can infer actions that typically lead to the desired reward. The second system, in blue, learns which actions should generally be chosen in a given state, and we refer to it as 'habit', because it suggests actions without considering the value of the reward currently available. Both goal-directed and habit systems propose an action, and their influence depends on their relative certainty.

**Figure 1B** gives an overview of how the systems mentioned above contribute to action planning, in a typical task. During initial trials, the valuation system (shown in red) evaluates the current state $s$ and computes the value of desired reward $v$, and the goal-directed system selects the action $a$. At this stage the habit system contributes little to the planning process as its uncertainty is high. As the training progresses, the habit system learns to mimic the choices made by the goal-directed system (**Miller et al., 2019**). On later trials the action is jointly determined by the habit and goal-directed systems (**Figure 1B**), and their relative contributions depend on their levels of certainty.

The details of the above computations in the framework will be described in the next section, and it will be later shown how an algorithm inferring action can be implemented in a network resembling the anatomy of the basal ganglia. But before going through a mathematical description, let us first provide an overview of this implementation (**Figure 1C**). In this implementation, the valuation, goal-directed and habit systems are mapped on the spectrum of cortico-basal ganglia loops (**Alexander et al., 1986**), ranging from valuation in a loop including ventral striatum, to habit in a loop including the dorsolateral striatum that has been shown to be critical for habitual behaviour (**Burton et al., 2015**). In the DopAct framework, the probability distributions learned by the actor are encoded in the strengths of synaptic connections in the corresponding loops, primarily in cortico-striatal connections. As in a standard implementation of the critic (**Houk et al., 1995**), the parameters of the value function learned by the valuation system are encoded in cortico-striatal connections of the corresponding loop.

Analogous to classical reinforcement learning theory, dopaminergic neurons play a critical role in learning, and encode errors in predictions made by the systems in the DopAct framework. However, by contrast to the standard theory, dopaminergic neurons do not all encode the same signal, but

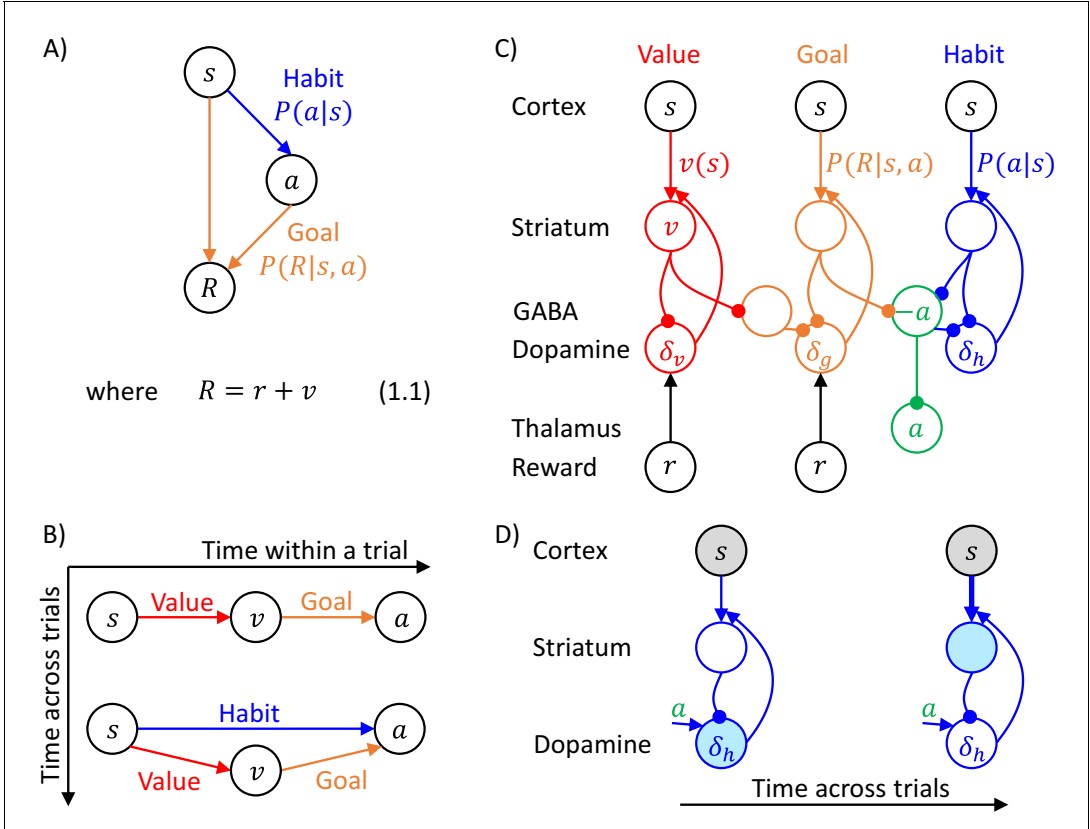

**Figure 1.** Overview of systems within the DopAct framework. (**A**) Probabilistic model learned by the actor. Random variables are indicated by circles, and arrows denote dependencies learned by different systems. (**B**) Schematic overview of information processing in the framework at different stages of task acquisition. (**C**) Mapping of the systems on different parts of the cortico-basal ganglia network. Circles correspond to neural populations located in the regions indicated by labels to the left, where 'Striatum' denotes medium spiny neurons expressing D1 receptors, 'GABA' denotes inhibitory neurons located in vicinity of dopaminergic neurons, and 'Reward' denotes neurons providing information on the magnitude of instantaneous reward. Arrows denote excitatory projections, while lines ending with circles denote inhibitory projections. (**D**) Schematic illustration of the mechanism of habit formation. Notation as in panel C, but additionally shading indicates the level of activity, and thickness of lines indicates the strength of synaptic connections.

instead dopaminergic populations in different systems compute errors in predictions made by their corresponding system. Since both valuation and goal-directed systems learn to predict reward, the dopaminergic neurons in these systems encode reward prediction errors (which slightly differ between these two systems, as will be illustrated in simulations presented later). By contrast, the habit system learns to predict action on the basis of a state, so its prediction error encodes how the currently chosen action differs from a habitual action in the given state. Thus these dopaminergic neurons respond to non-habitual actions in the DopAct framework. We denote the prediction errors in the valuation, goal-directed and habit systems by $\delta_v$, $\delta_g$ and $\delta_h$, respectively. The dopaminergic neurons send these prediction errors to the striatum, where they trigger plasticity of cortico-striatal connections.

In the DopAct framework, habits are formed through a process in which the habit system learns to mimic the goal-directed system. Unlike in a previous model of habit formation (*Daw et al., 2005*), in the DopAct framework learning in the habit system is not driven by a reward prediction error, but by a signal encoding a difference between chosen and habitual actions. At the start of training, when an action is selected mostly by the goal-directed system, the dopaminergic neurons in the habit system receive an input encoding the chosen action, but the striatal neurons in the habit system are not yet able to predict this action, resulting in a prediction error encoded in dopaminergic activity (left display in *Figure 1D*). This prediction error triggers plasticity in the striatal neurons of the habit system, so they tend to predict this action in the future (right display in *Figure 1D*).

The systems communicate through an 'ascending spiral' structure of striato-dopaminergic projections identified by *Haber et al., 2000*. These Authors observed that dopaminergic neurons within a given loop project to the corresponding striatal neurons, while the striatal neurons project to the dopaminergic neurons in the corresponding and next loops, and they proposed that the projections to the next loop go via interneurons, so they are effectively excitatory (*Figure 1C*). In the DopAct framework, once the striatal neurons in the valuation system compute the value of the state $v$, they send it to the dopaminergic neurons in the goal-directed system.

In the DopAct framework, dopamine in the goal-directed system plays a role in both action planning and learning, and now an overview of this role is given. In agreement with classical reinforcement learning theory, the dopaminergic activity $\delta_g$ encodes reward prediction error, namely the difference between the reward $R$ (including both obtained and available reward) and the expected reward (*Schultz et al., 1997*), but in the DopAct framework the expectation of reward in the goal-directed system is computed on the basis of the current action plan. Therefore, this reward expectation only arises from formulating a plan to achieve it. Consequently, when a reward is available, the prediction error $\delta_g$ can only be reduced to zero, once a plan to obtain the reward is formulated.

To gain an intuition for how the goal-directed system operates, let us consider a simple example of a hungry rat in a standard operant conditioning experiment. Assume that the rat has been trained that after pressing a lever a food pellet is delivered (*Figure 2A*). Consider a situation in which a lever is suddenly made available to the animal. Its sight allows the valuation system to predict that reward is available, and it sends an estimated value of the reward to the goal-directed system. Such input induces a reward prediction error in the goal-directed system, because this system has received information that a reward is available, but has not yet prepared actions to obtain the reward, hence it does not expect any reward for its action. The resulting prediction error triggers a process of planning actions that can get the reward. This facilitation of planning arises in the network, because the dopaminergic neurons in the goal-directed system project to striatal neurons (*Figure 1C*), and increase their excitability. Once an appropriate action has been computed, the animal starts to expect the available reward, and the dopamine level encoding the prediction error decreases. Importantly, in this network dopamine provides a crucial feedback to striatal neurons on whether the current action plan is sufficient to obtain the available reward. If it is not, this feedback triggers changes in the action plan until it becomes appropriate. Thus the framework suggests why it is useful for the neurons encoding reward prediction error to be involved in planning, namely it suggests that this prediction error provides a useful feedback for the action planning system, informing if the plan is suitable to obtain the reward.

It is worth explaining why the reward expectation in the goal-directed system arises already once an action is computed and before it is implemented. It happens in the DopAct framework, because

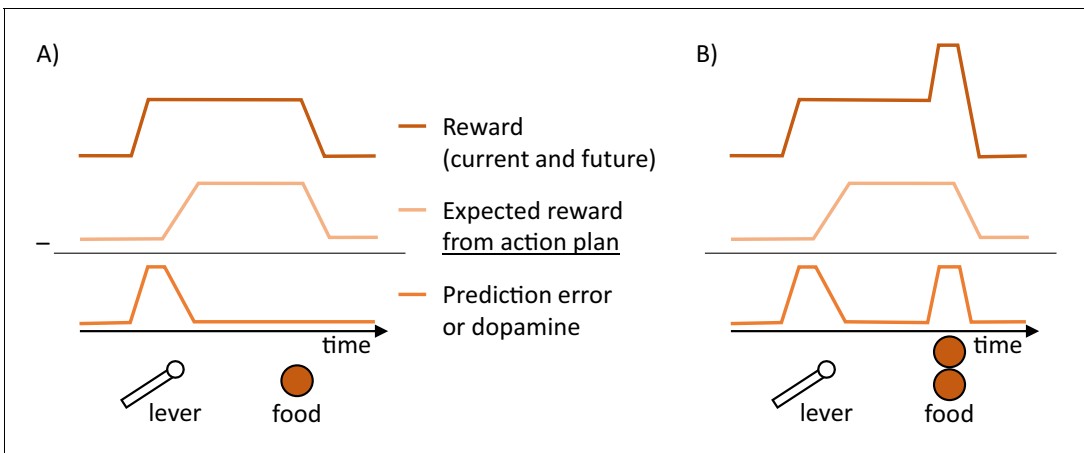

**Figure 2.** Schematic illustration of changes in dopaminergic activity in the goal-directed system while a hungry rat presses a lever and a food pellet is delivered. (**A**) Prediction error reduced by action planning. The prediction error encoded in dopamine (bottom trace) is equal to a difference between the reward available (top trace) and the expectation of reward arising from a plan to obtain it (middle trace). (**B**) Prediction errors reduced by both action planning and learning.

the striatal neurons in the goal-directed system learn over trials to predict that particular pattern of activity of neurons encoding action in the basal ganglia (which subsequently triggers a motor response) leads to reward in the future. This mechanism is fully analogous to that in the temporal-difference learning model used to describe classical conditioning, where the reward expectation also arises already after a stimulus, because the striatal neurons learn that the pattern of cortical inputs to the basal ganglia encoding the state (i.e. the stimulus) will lead to a reward (*Schultz et al., 1997*). In the goal-directed system of DopAct, an analogous reward prediction is made, but not only on the basis of a state, but on the basis of a combination of state and action.

The prediction error in the goal-directed system also allows the animal to learn about the rewards resulting from actions. In the example we considered above such learning would be necessary if the amount of reward changed, for example to two pellets (*Figure 2B*). On the first trial after such change, a prediction error will be produced after reward delivery. This prediction error can be reduced by learning, so the animal will expect such increased reward in the future trials and no longer produce prediction error at reward delivery. In summary, the prediction errors in the goal-directed system are reduced by both planning and learning, as in active inference (*Friston, 2010*). Namely, the prediction errors arising from rewards becoming available are reduced within trials by formulating plans to obtain them, and the prediction errors due to outcomes of actions differing from expectations are reduced across trials by changing weights of synaptic connection encoding expected reward.

The next three sections will provide the details of the DopAct framework. For clarity, we will follow Marr's levels of description, and discuss computations, an algorithm, and its implementation in the basal ganglia network.

## Computations during planning and learning

To illustrate the computations in the framework we will consider a simple task, in which only an intensity of a single action needs to be chosen. Such choice has to be made by animals in classical experiments investigating habit formation, where the animals are offered a single lever, and need to decide how frequently to press it. Furthermore, action intensity often needs to be chosen by animals also in the wild (e.g. a tiger deciding how vigorously pounce on a prey, a chimpanzee choosing how strongly hit a nut with a stone, or a sheep selecting how quickly eat the grass). Let us denote the action intensity by $a$. Let us assume that the animal chooses it on the basis of the reward it expects $R$ and the stimulus $s$ (e.g. the size of prey, nut or grass). Thus the animal needs to infer an action intensity sufficient to obtain the desired reward (but not larger to avoid unnecessary effort).

Let us consider the computation in the DopAct framework during action planning. During planning, the animal has not received any reward yet $r = 0$, so according to Equation 1.1, the total reward is equal to the reward available $R = v$. While planning to obtain this reward, the actor combines information from the goal-directed system (encoding how the reward depends on actions taken in given states), and the habit system (encoding the probability distribution of generally selecting actions in particular states). These two pieces information are combined according to Bayes' theorem (Equation 3.1 in *Figure 3*), which states that the posterior probability of selecting a particular action given available reward is proportional to the product of a likelihood of the reward given the action, which we propose is represented in the goal-directed system, and a prior, which we propose is encoded by the habit system.

In the DopAct framework, an action $a$ is selected which maximizes the probability $P(a|R, s)$. An analogous way of selecting actions has been used in models treating planning as inference (*Attias, 2003*), and it has been nicely summarized by *Solway and Botvinick, 2012* 'The decision process takes the occurrence of reward as a premise, and leverages the generative model to determine which course of action best explains the observation of reward.' In this paper, we make explicit the rationale for this approach: The desired amount of resources that should be acquired depends on the levels of reserves (and a given state); this value is computed by the valuation system, and the actor needs to find the action depending on this reward. Let us provide a further rationale for selecting an action $a$ which maximizes $P(a|R, s)$, by analysing what this probability expresses. Let us consider the following hypothetical scenario: An animal selected an action without considering the desired reward, that is by sampling it from its default policy $P(a|s)$ provided by the habit system, and obtained reward $R$. In this case, $P(a|R, s)$ is the probability that the selected action was $a$. When an animal knows the amount of resource desired $R$, then instead of just relying on the prior, the animal

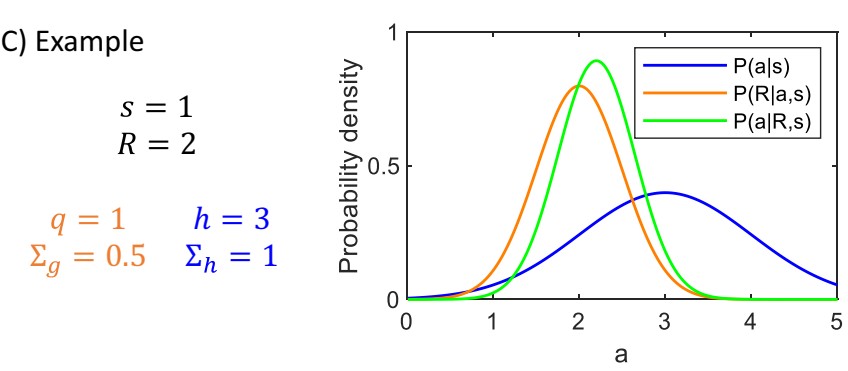

A) Computation

Planning ($R = v$): Choose action that maximizes

$$P(a|R,s) = \frac{\overbrace{P(R|a,s)}^{\text{Goal}}\overbrace{P(a|s)}^{\text{Habit}}}{P(R|s)} \qquad (3.1)$$

Learning ($R = r$): Update parameters to increase $P(R|s)$

B) Distributions

$$P(R|a,s) = f\big(R; aqs, \Sigma_g\big) \qquad P(a|s) = f(a; hs, \Sigma_h) \qquad (3.2)$$

$$\text{where} \quad f(x; \mu, \Sigma) = \frac{1}{\sqrt{2\pi\Sigma}} \exp\left(-\frac{(x-\mu)^2}{2\Sigma}\right) \qquad (3.3)$$

C) Example

$$s = 1$$
$$R = 2$$

$$q = 1 \qquad h = 3$$
$$\Sigma_g = 0.5 \qquad \Sigma_h = 1$$

**Figure 3.** Computational level. (**A**) Summary of computations performed by the actor. (**B**) Sample form of probability distributions. (**C**) An example of inference of action intensity. In this example the stimulus intensity is equal to $s = 1$, the valuation system computes desired reward $R = 2$, and the parameters of the probability distributions encoded in the goal-directed and habit systems are listed in the panel. The blue curve shows the distribution of action intensity, which the habit system has learned to be generally suitable for this stimulus. The orange curve shows probability density of obtaining reward of 2 for a given action intensity, and this probability is estimated by the goal-directed system. For the chosen parameters, it is the probability of obtaining 2 from a normal distribution with mean $a$. Finally, the green curve shows a posterior distribution computed from Equation 3.1.

should rather choose an action maximizing $P(a|R,s)$, which was the action most likely to yield this reward in the above scenario.

One may ask why it is useful to employ the habit system, instead of exclusively relying on the goal-directed system that encodes the relationship between rewards and actions. It is because there may be uncertainty in the action suggested by the goal-directed system, arising for example, from noise in the computations of the valuation system or inaccurate estimates of the parameters of the goal-directed system. According to Bayesian philosophy, in face of such uncertainty, it is useful to additionally bias the action by a prior, which here is provided by the habit system. This prior encodes an action policy that has overall worked in the situations previously experienced by the animal, so it is a useful policy to consider under the uncertainty in the goal-directed system.

To make the above computation more concrete, we need to specify the form of the prior and likelihood distributions. We first provide them for the example of choosing action intensity. They are given in *Figure 3B*, where $f(x; \mu, \Sigma)$ denotes the probability density of a normal distribution with

mean $\mu$ and variance $\Sigma$. In a case of the prior, we assume that action intensity is normally distributed around a mean given by stimulus intensity scaled by parameter $h$, reflecting an assumption that a typical action intensity often depends on a stimulus (e.g. the larger a nut, the harder a chimpanzee must hit it). On the other hand, in a case of the probability of reward $R$ maintained by the goal-directed system, the mean of the reward is equal to a product of action intensity and the stimulus size, scaled by parameter $q$. We assume that the mean reward depends on a product of $a$ and $s$ for three reasons. First, in many situations reward depends jointly on the size of the stimulus, and the intensity with which the action is taken, because if the action is too weak, the reward may not be obtained (e.g. a prey may escape or a nut may not crack), and the product captures this dependence of reward on a conjunction of stimulus and action. Second, in many foraging situations, the reward that can be obtained within a period of time is proportional to a product of $a$ and $s$ (e.g. the amount of grass eaten by a sheep is proportional to both how quickly the sheep eats it, and how high the grass is). Third, when the framework is generalized to multiple actions later in the paper, the assumption of reward being proportional to a product of $a$ and $s$ will highlight a link with classical reinforcement learning. We denote the variances of the distributions of the goal-directed and habit systems by $\Sigma_g$ and $\Sigma_h$. The variance $\Sigma_g$ quantifies to what extent the obtained rewards have differed from those predicted by the goal-directed system, while the variance $\Sigma_h$ describes by how much the chosen actions have differed from the habitual actions.

*Figure 3C* shows an example of probability distributions encoded by the two systems for sample parameters. It also shows a posterior distribution $P(a|R, s)$, and please note that its peak is in between the peaks of the distributions of the two systems, but it is closer to the peak of a system with smaller uncertainty (orange distribution is narrower). This illustrates how in the DopAct framework, the action is inferred by incorporating information from both systems, but weighting it by the certainty of the systems.

In addition to action planning, the animal needs to learn from the outcomes, to predict rewards more accurately in the future. After observing an outcome, the valuation system no longer predicts future reward $v = 0$, so according to Equation 1.1 the total reward is equal to the reward actually obtained $R = r$. The parameters of the distributions should be updated to increase $P(R|s)$, so in the future the animal is less surprised by the reward obtained in that state (*Figure 3A*).

## Algorithm for planning and learning

Let us describe an algorithm used by the actor to infer action intensity $a$ that maximizes the posterior probability $P(a|R, s)$. This posterior probability could be computed from Equation 3.1, but note that $a$ does not occur in the denominator of that equation, so we can simply find the action that maximizes the numerator. Hence, we define an objective function $F$ equal to a logarithm of the numerator of Bayes' theorem (Equation 4.1 in *Figure 4*). Introducing the logarithm will simplify function $F$ because it will cancel with exponents present in the definition of normal density (Equation 3.3), and it does not change the position of the maximum of the numerator because the logarithm is a monotonic function. For example, the green curve in *Figure 4B* shows function $F$ corresponding to the posterior probability in *Figure 3C*. Both green curves have the maximum at the same point, so instead of searching for a maximum of a posterior probability, we can seek the maximum of a simpler function $F$.

During action planning the total reward is equal to reward available, so we set $R = v$ in Equation 4.1, and we find the action maximizing $F$. This can be achieved by initializing $a$ to any value, and then changing it proportionally to the gradient of $F$ (Equation 4.2). *Figure 4B* illustrates that with such dynamics, the value of $a$ approaches a maximum of $F$. Once $a$ converges, the animal may select the action with the corresponding intensity. In summary, this method yields a differential equation describing an evolution of a variable $a$, which converges to a value of $a$ that maximizes $P(a|R, s)$.

After obtaining a reward, $R$ is equal to the reward obtained, so we set $R = r$ in Equation 4.1, and the values of parameters are changed proportionally to the gradient of $F$ (Equations 4.3). Such parameter updates allow the model to be less surprised by the rewards (as aimed for in *Figure 3A*), because under certain assumptions function $F$ expresses 'negative free energy'. The negative free energy (for the inference problem considered in this paper) is defined as $F = \ln P(R|s) - KL$, where $KL$ is the Kullback-Leibler divergence between $P(a|R, s)$ and an estimate of this distribution (a detailed definition and an explanation for why $F$ given in Equation 4.1 expresses negative free energy for an analogous problem is given by *Bogacz, 2017*). Importantly, since $KL \geq 0$, the negative

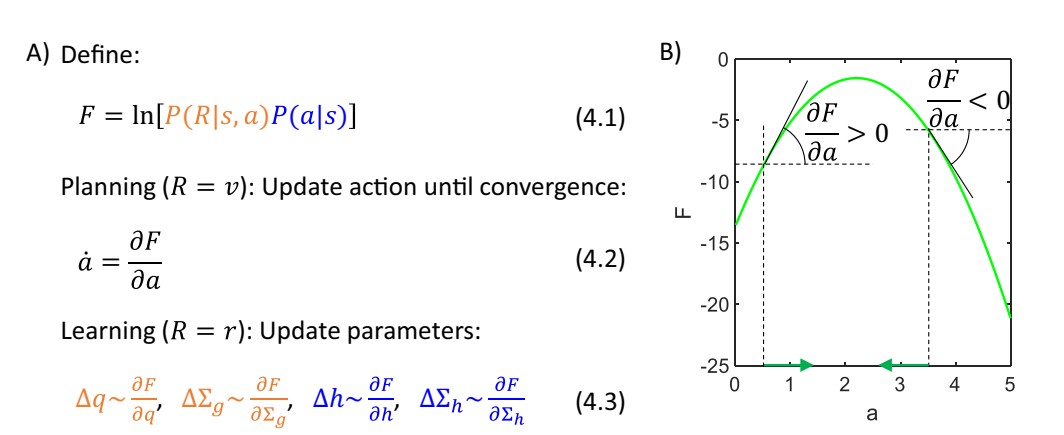

**Figure 4.** Algorithmic level. (**A**) Summary of the algorithm used by the actor. (**B**) Identifying an action based on a gradient of $F$. The panel shows an example of a dependence of $F$ on $a$, and we wish $a$ to take the value maximizing $F$. To find the action, we let $a$ to change over time in proportion to the gradient of $F$ over $a$ (Equation 4.2, where the dot over $a$ denotes derivative over time). For example, if the action is initialized to $a = 1.5$, then the gradient of $F$ at this point is positive, so $a$ is increased (**Equation 4.2**), as indicated by a green arrow on the x-axis. These changes in $a$ continue until the gradient is no longer positive, i.e. when $a$ is at the maximum. Analogously, if the action is initialized to $a = 3.5$, then the gradient of $F$ is negative, so $a$ is decreased until it reaches the maximum of $F$.

free energy provides a lower bound on $P(R|s)$ (**Friston, 2005**). Thus changing the parameters to increase $F$, rises the lower bound on $P(R|s)$, and so it tends to increase $P(R|s)$.

Let us derive the details of the algorithm (general form of which is given in **Figure 4A**) for the problem of choosing action intensity. Let us start with considering a special case in which both variance parameters are fixed to $\Sigma_g = \Sigma_h = 1$, because then the form of the algorithm and its mapping on the network are particularly beautiful. Substituting probability densities of likelihood and prior distributions (Equations 3.2-3.3) for the case of unit variances into Equation 4.1 (and ignoring

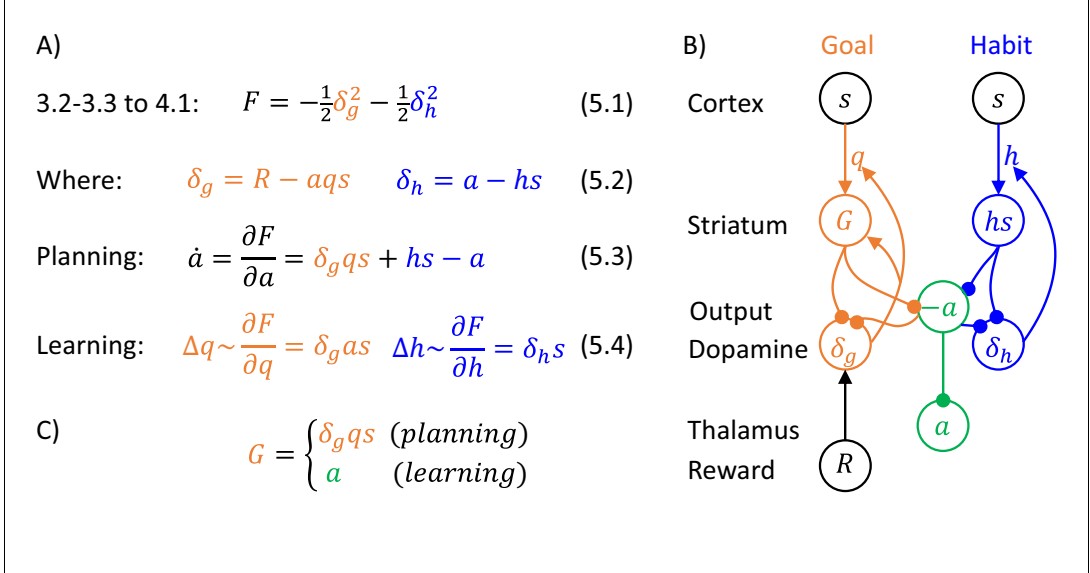

**Figure 5.** Description of a model selecting action intensity, in a case of unit variances. (**A**) Details of the algorithm. (**B**) Mapping of the algorithm on network architecture. Notation as in **Figure 1C**, and additionally 'Output' denotes the output nuclei of the basal ganglia. (**C**) Definition of striatal activity in the goal-directed system.

constants $1/\sqrt{2\pi}$), we obtain the expression for the objective function $F$ in Equation 5.1 (*Figure 5A*). We see that $F$ consists of two terms, which are the squared prediction errors associated with goal-directed and habit systems. The prediction error for the goal-directed system describes how the reward differs from the expected mean, while the prediction error of the habit system expresses how the chosen action differs from that typically chosen in the current state (Equations 5.2). As described in the previous section, action intensity can be found by changing its value according to a gradient of $F$ (Equation 4.2). Computing the derivative of $F$ over $a$, we obtain Equation 5.3, where the two colours indicate terms connected with derivatives of the corresponding prediction errors. Finally, when the reward is obtained, we modify the parameters proportionally to the derivatives of $F$ over the parameters, which are equal to relatively simple expressions in Equations 5.4.

*Figure 5A* illustrates the key feature of the DopAct framework, that both action planning and learning can be described by the same process. Namely in both planning and learning, certain variables (the action intensity and synaptic weights, respectively) are changed to maximize the same function $F$ (Equations 5.3 and 5.4). Since $F$ is a negative of the sum of prediction errors (Equation 5.1), both action planning and learning are aimed at reducing prediction errors.

## Network selecting action intensity

The key elements of the algorithm in *Figure 5A* naturally map on the known anatomy of striato-dopaminergic connections. This mapping relies on three assumptions analogous to those typically made in models of the basal ganglia: (i) the information about state $s$ is provided to the striatum by cortical input, (ii) the parameters of the systems $q$ and $h$ are encoded in the cortico-striatal weights, and (iii) the computed action intensity is represented in the thalamus (*Figure 5B*). Under these assumptions, Equation 5.3 describing an update of action intensity can be mapped on the circuit: The action intensity in the model is jointly determined by the striatal neurons in the goal-directed and habit systems, which compute the corresponding terms of Equation 5.3, and communicate them by projecting to the thalamus via the output nuclei of the basal ganglia. The first term $\delta_g q s$ can be provided by striatal neurons in the goal-directed system (denoted by $G$ in *Figure 5B*): They receive cortical input encoding stimulus intensity $s$, which is scaled by cortico-striatal weights encoding parameter $q$, so these neurons receive synaptic input $qs$. To compute $\delta_g q s$, the gain of the striatal neurons in the goal-directed system needs to be modulated by dopaminergic neurons encoding prediction error $\delta_g$ (this modulation is represented in *Figure 5B* by an arrow from dopaminergic to striatal neurons). Hence, these dopaminergic neurons drive an increase in action intensity until the prediction error they represent is reduced (as discussed in *Figure 2*). The second term $hs$ in Equation 5.3 can be computed by a population of neurons in the habit system receiving cortical input via connection with the weight $h$. Finally, the last term $-a$ simply corresponds to a decay.

In the DopAct framework, dopaminergic neurons within each system compute errors in the predictions about the corresponding variable, i.e. reward for the goal-directed system, and action for the habit system. Importantly, in the network on *Figure 5B* this computation can be performed locally, i.e. the dopaminergic neurons receive inputs encoding all quantities necessary to compute their corresponding errors. In the habit system, the prediction error is equal to a difference between action $a$ and expectation $hs$ (blue Equation 5.2). Such error can be easily computed in a network of *Figure 5B*, where the dopaminergic neurons in the habit system receive effective input form the output nuclei equal to $a$ (as they receive inhibition equal to $-a$), and inhibition $hs$ from the striatal neurons. In the goal-directed system, the expression for prediction error is more complex (orange Equation 5.2), but importantly, all terms occurring in the equation could be provided to dopaminergic neurons in the goal-directed system via connections shown in *Figure 5B* ($qs$ could be provided by the striatum, while $a$ thorough an input from the output nuclei which have been reported to project to dopaminergic neurons [*Watabe-Uchida et al., 2012*]).

Once the actual reward is obtained, changing parameters proportionally to prediction errors (Equations 5.4) can arise due to dopaminergic modulation of the plasticity of cortico-striatal connections (represented in *Figure 5B* by arrows going from dopamine neurons to parameters). With such a modulation, learning could be achieved through local synaptic plasticity: The update of a weight encoding parameter $h$ (blue Equation 5.4) is simply proportional to the product of presynaptic ($s$) and dopaminergic activity ($\delta_h$). In the goal-directed system, orange Equation 5.4 corresponds to local plasticity, if at the time of reward the striatal neurons encode information about action intensity (see

definition of $G$ in *Figure 5C*). Such information could be provided from the thalamus during action execution. Then the update of synaptic weight encoding parameter $q$ will correspond to a standard three-factor rule (*Kuśmierz et al., 2017*) involving a product of presynaptic ($s$), postsynaptic ($a$) and dopaminergic activity ($\delta_g$).

The model can be extended so that the parameters $\Sigma_g$ and $\Sigma_h$ describing variances of distributions are encoded in synaptic connections or internal properties of the neurons (e.g. leak conductance). In such an extended model, the action proposals of the two systems are weighted according to their certainties. *Figure 6A* shows the general description of the algorithms which is analogous to that in *Figure 5A*. The action intensity is driven by both goal-directed and habit systems, but now their contributions are normalised by the variance parameters. For the habit system this normalization is stated explicitly in Equation 6.2, while for the goal-directed system it comes from a normalization of prediction error by variance in orange Equation 6.3 (it is not necessary to normalize habit prediction error by variance because the contribution of the habit system is already normalized in Equation 6.2).

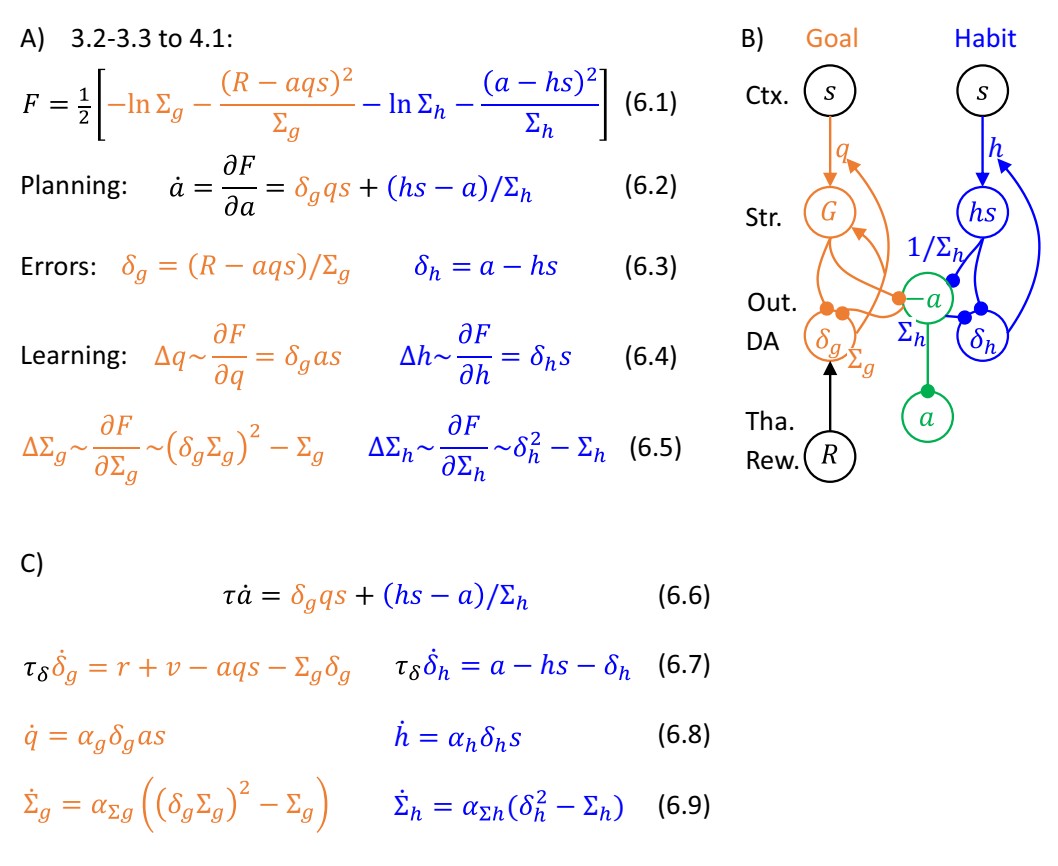

A) 3.2-3.3 to 4.1:

$$F = \tfrac{1}{2}\left[-\ln\Sigma_g - \frac{(R-aqs)^2}{\Sigma_g} - \ln\Sigma_h - \frac{(a-hs)^2}{\Sigma_h}\right] \quad (6.1)$$

Planning: $\quad \dot{a} = \dfrac{\partial F}{\partial a} = \delta_g qs + (hs-a)/\Sigma_h \quad (6.2)$

Errors: $\quad \delta_g = (R-aqs)/\Sigma_g \qquad \delta_h = a - hs \quad (6.3)$

Learning: $\quad \Delta q \sim \dfrac{\partial F}{\partial q} = \delta_g as \qquad \Delta h \sim \dfrac{\partial F}{\partial h} = \delta_h s \quad (6.4)$

$\Delta\Sigma_g \sim \dfrac{\partial F}{\partial \Sigma_g} \sim (\delta_g\Sigma_g)^2 - \Sigma_g \qquad \Delta\Sigma_h \sim \dfrac{\partial F}{\partial \Sigma_h} \sim \delta_h^2 - \Sigma_h \quad (6.5)$

C)

$$\tau\dot{a} = \delta_g qs + (hs-a)/\Sigma_h \quad (6.6)$$

$\tau_\delta\dot{\delta}_g = r + v - aqs - \Sigma_g\delta_g \qquad \tau_\delta\dot{\delta}_h = a - hs - \delta_h \quad (6.7)$

$\dot{q} = \alpha_g\delta_g as \qquad\qquad \dot{h} = \alpha_h\delta_h s \quad (6.8)$

$\dot{\Sigma}_g = \alpha_{\Sigma g}\left((\delta_g\Sigma_g)^2 - \Sigma_g\right) \qquad \dot{\Sigma}_h = \alpha_{\Sigma h}(\delta_h^2 - \Sigma_h) \quad (6.9)$

**Figure 6.** Description of a model selecting action intensity. (**A**) Details of the algorithm. The update rules for the variance parameters can be obtained by computing derivatives of $F$, giving $\delta_g^2 - 1/\Sigma_g$ and $\delta_h^2/\Sigma_h^2 - 1/\Sigma_h$, but to simplify these expressions, we scale them by $\Sigma_g^2$ and $\Sigma_h^2$, resulting in Equations 6.5. Such scaling does not change the value to which the variance parameters converge because $\Sigma_g^2$ and $\Sigma_h^2$ are positive. (**B**) Mapping of the algorithm on network architecture. Notation as in *Figure 5B*. This network is very similar to that shown in *Figure 5B*, but now the projection to output nuclei from the habit system is weighted by its precision $1/\Sigma_h$ (to reflect the weighting factor in Equation 6.2), and also the rate of decay (or relaxation to baseline) in the output nuclei needs to depend on $\Sigma_h$. One way to ensure that the prediction error in goal-directed system is scaled by $\Sigma_g$ is to encode $\Sigma_g$ in the rate of decay or leak of these prediction error neurons (*Bogacz, 2017*). Such decay is included as the last term in orange Equation 6.7 describing the dynamics of prediction error neurons. Prediction error evolving according to this equation converges to the value in orange Equation 6.3 (the value in equilibrium can be found by setting the left hand side of orange Equation 6.7 to 0, and solving for $\delta_g$). In Equation 6.7, total reward $R$ was replaced according to Equation 1.1 by the sum of instantaneous reward $r$, and available reward $v$ computed by the valuation system. (**C**) Dynamics of the model.

There are several ways of including the variance parameters in the network, and one of them is illustrated in *Figure 6B* (see caption for details). The updates of the variance parameters (Equations 6.5) only depend on the corresponding prediction errors and the variance parameters themselves, so they could be implemented with local plasticity, if the neurons encoding variance parameters received corresponding prediction errors. *Figure 6C* provides a complete description of the dynamics of the simulated model. It parallels that in *Figure 6B*, but now explicitly includes time constants for update of neural activity ($\tau$, $\tau_\delta$), and learning rates for synaptic weights ($\alpha$ with corresponding indices).

As described in the Materials and methods, a simple model of the valuation system based on standard temporal-difference learning was employed in simulations (because the simulations corresponded to a case of low level of animal's reserves). Striatal neurons in the valuation system compute the reward expected in a current state on the basis of parameters $w_t$ denoting estimates of reward at time $t$ after a stimulus, and following standard reinforcement learning we assume that these parameters are encoded in cortico-striatal weights. The dopaminergic neurons in the valuation system encode the prediction error similar to that in the temporal-difference learning model, and after reward delivery, they modulate plasticity of cortico-striatal connections. The Method section also provides details of the implementation and simulations of the model.

## Simulations of action intensity selection

To illustrate how the model mechanistically operates and to help relate it to experimental data, we now describe a simulation of the model inferring action intensity. On each simulated trial the model selected action intensity, after observing a stimulus, which was set to $s = 1$. The reward obtained depended on action intensity as shown in *Figure 7A*, according to $r = 5 \tanh(3a/5) - a$. Thus, the reward was proportional to the action intensity, transformed through a saturating function, and a cost was subtracted proportional to the action intensity, that could correspond to a price for making an effort. We also added Gaussian noise to reward (with standard deviation $\sigma_r = 0.5$) to account for randomness in the environment, and to action intensity to account for imprecision of the motor system or exploration.

*Figure 7AB* shows how the quantities encoded in the valuation system changed throughout the learning process. The pattern of prediction errors in this figure is very similar to that expected from the temporal difference model, as the valuation system was based on that model. The stimulus was presented at time $t = 1$. On the first trial (left display) the simulated animal received a positive reward at time $t = 2$ (dashed black curve) due to stochastic nature of the rewards in the simulation. As initially the expectation of reward was low (dashed red curve), the reward triggered a substantial prediction error (solid red curve). The middle and right plots show the same quantities after learning. Now the prediction error was produced after the presentation of the stimulus, because after seeing the stimulus a simulated animal expected more reward than before the stimulus. In the middle display the reward received at time $t = 2$ was very close to the expectation, so the prediction error at the time of the reward was close to 0. In the right display the reward happened to be lower than usual (due to noise in the reward), which resulted in a negative prediction error. Note that the pattern of prediction errors in the valuation system in *Figure 7B* resembles the famous figure showing the activity of dopaminergic neurons during conditioning (*Schultz et al., 1997*).

*Figure 7C* shows the prediction errors in the actor and action intensity on the same trials that were visualised in *Figure 7B*. Prediction errors in the goal-directed system follow a similar pattern as in the valuation system in the left and middle displays in *Figure 7C*, that is before the behaviour becomes habitual. The middle display in *Figure 7C* shows simulated neural activity that was schematically illustrated in *Figure 2A*: As the valuation system detected that a reward was available (see panel above), it initially resulted in a prediction error in the goal-directed system, visible as an increase in the orange curve. This prediction error triggered a process of action planning, so with time the green curve representing planned action intensity increased. Once the action plan has been formulated, it provided a reward expectation, so the orange prediction error decreased. When an action became habitual after extensive training (right display in *Figure 7C*), the prediction error in the goal-directed system started to qualitatively differ from that in the valuation system. At this stage of training, the action was rapidly computed by the habit system, and the goal-directed system was too slow to lead action planning, so the orange prediction error was lower. This illustrates

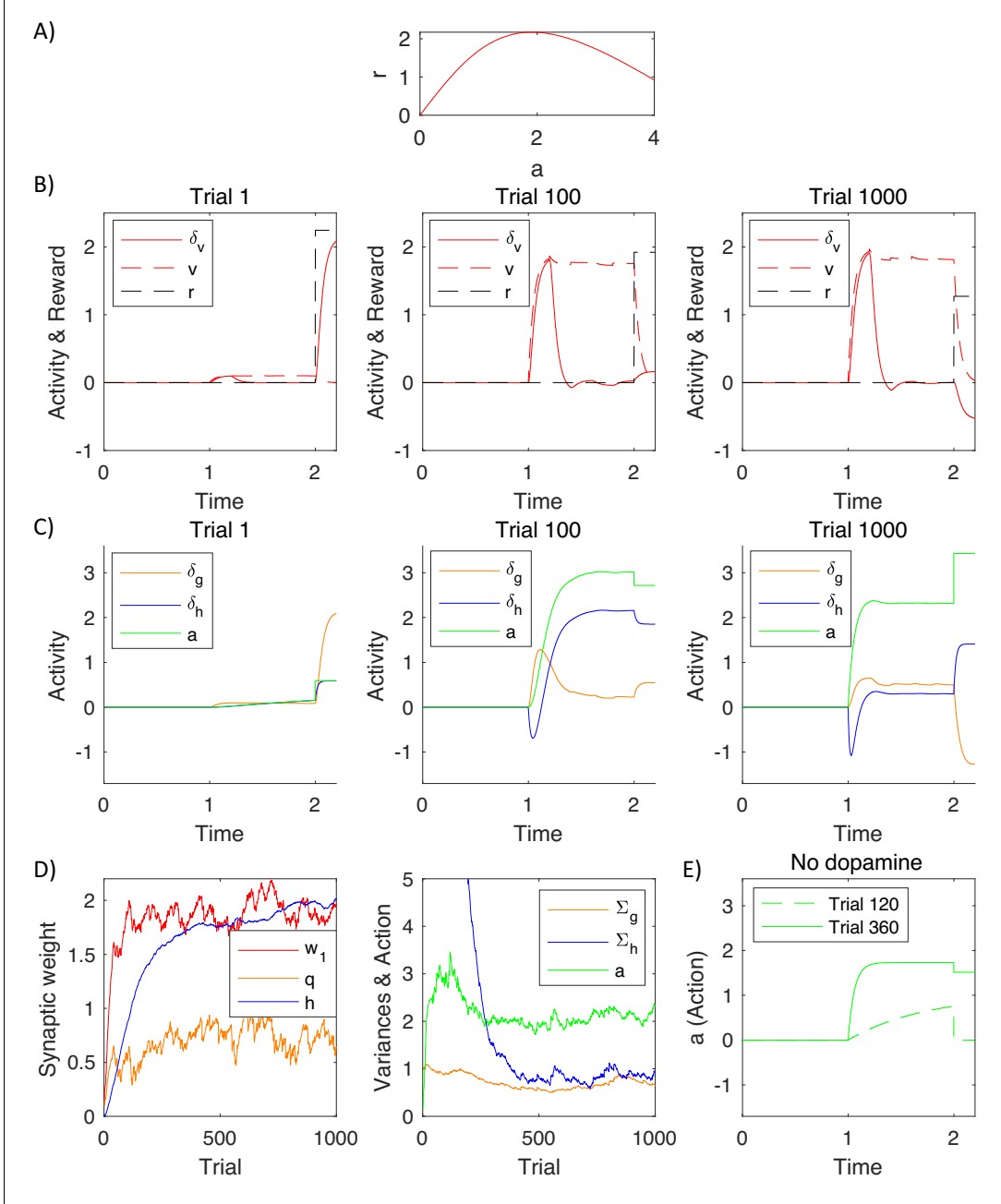

**Figure 7.** Simulation of a model selecting action intensity. (**A**) Mean reward given to a simulated agent as a function of action intensity. (**B**) Changes in variables encoded by the valuation system in different stages of task acquisition. (**C**) Changes in variables encoded by the actor. (**D**) Changes in model parameters across trials. The green curve in the right display shows the action intensity at the end of a planning phase of each trial. (**E**) Action intensity inferred by the model after simulated blocking of dopaminergic neurons.

that in the DopAct framework reward expectations in the goal-directed system can arise even if an action is computed by the habit system.

The prediction error in the habit system follows a very different pattern than in other systems. Before an action became habitual, the prediction errors in the habit system arose after the action has been computed (middle display in *Figure 7C*). Since the habit system has not formed significant habits on early trials, it was surprised by the action, and this high value of blue prediction error drove its learning over trials. Once the habit system was highly trained (right display in *Figure 7C*) it rapidly drove action planning, so the green curve showing planned action intensity increased more rapidly.

Nevertheless, due to the dynamics in the model, the increase in action intensity was not instant, so there was a transient negative prediction error in the habit system while an action was not yet equal to the intensity predicted by the habit system. The prediction error in the habit system at the time of action execution depended on how the chosen action differed from a habitual one, rather than on the received reward (e.g. in the right display in *Figure 7C*, $\delta_h > 0$ because the executed action was stronger than the planned one due to motor noise, despite reward being lower than expected).

*Figure 7D* shows how the parameters in the model evolved over the trials in the simulation. The left display shows changes in the parameters of the three systems. A parameter of the valuation system correctly converged to the maximum value of the reward available in the task $w_1 \approx 2$ (i.e. the maximum of the curve in *Figure 7A*). The parameter of the habit system correctly converged to $h \approx 2$, i.e. typical action intensity chosen over trials (shown by a green curve in the right display of *Figure 7D*). The parameter of the goal-directed system converged to a vicinity of $q \approx 1$, which allows the goal-directed system to expect the reward of 2 after selecting an action with intensity 2 (according to orange Equation 3.2 the reward expected by the goal-directed system is equal to $aqs \approx 2 \times 1 \times 1 = 2$). The right display in *Figure 7D* shows how the variance parameters in the goal-directed and habit systems changed during the simulation. The variance of the habit system was initialised to a high value, and it decreased over time, resulting in an increased certainty of the habit system.

Dopaminergic neurons in the model are only required to facilitate planning in the goal-directed system, where they increase excitability of striatal neurons, but not in the habit system. To illustrate it, *Figure 7E* shows simulations of a complete dopamine depletion in the model. It shows action intensity produced by the model in which following training, all dopaminergic neurons were set to 0. After 119 trials of training, on the 120th trial, the model was unable to plan an action. By contrast, after 359 training trials (when the uncertainty of the habit system has decreased – see the blue curve in right display of *Figure 7D*), the model was still able to produce a habitual response, because dopaminergic neurons are not required for generating habitual responses in the model. This parallels the experimentally observed robustness of habitual responses to blocking dopaminergic modulation (*Choi et al., 2005*).

## Simulations of effects observed in conditioning experiments

This section shows that the model is able to reproduce two key patterns of behaviour that are thought to arise from interactions between different learning systems, namely the resistance of habitual responses to reward devaluation (*Dickinson, 1985*), and Pavlovian-instrumental transfer (*Estes, 1943*).

In experiments investigating devaluation, animals are trained to press a level (typically multiple times) for reward, for example food. Following this training the reward is devalued in a subgroup of animals, e.g. the animals in the devaluation group are fed to satiety, so they no longer desire the reward. Top displays in *Figure 8A* replot experimental data from one such study (*Dickinson et al., 1995*). The displays show the average number of lever presses made by trained animals during a testing period in which no reward was given for lever pressing. The dashed and solid curves correspond to devaluation and control groups, and the two displays correspond to groups of animals trained for different periods, that is trained until they received 120 or 360 rewards respectively. *Figure 8A* illustrates two key effects. First, all animals eventually reduced lever pressing with time, thus demonstrating extinction of the previously learned responses. Second, the effect of devaluation on initial testing trials depended on the amount of training. In particular, in the case of animals that received moderate amount of training (top left display) the number of responses in the first bin was much lower for the devaluation group than control group. By contrast, highly trained animals (top right display) produced more similar numbers of responses in the first bin irrespective of devaluation. Such production of actions despite their consequence being no longer desired is considered as a hallmark of habit formation.

The model can also produce insensitivity to devaluation with extensive training. Although the experimental tasks involving pressing levers multiple times is not identical to choosing intensity of a single action, such tasks could be conceptualized as a choice of the frequency of pressing a lever, that could also be described by a single number $a$. Furthermore, the average reward rate experienced by an animal in paradigms typically used in studies of habit formation (variable interval schedules that will be explained in Discussion) may correspond to a non-monotonic function similar to that

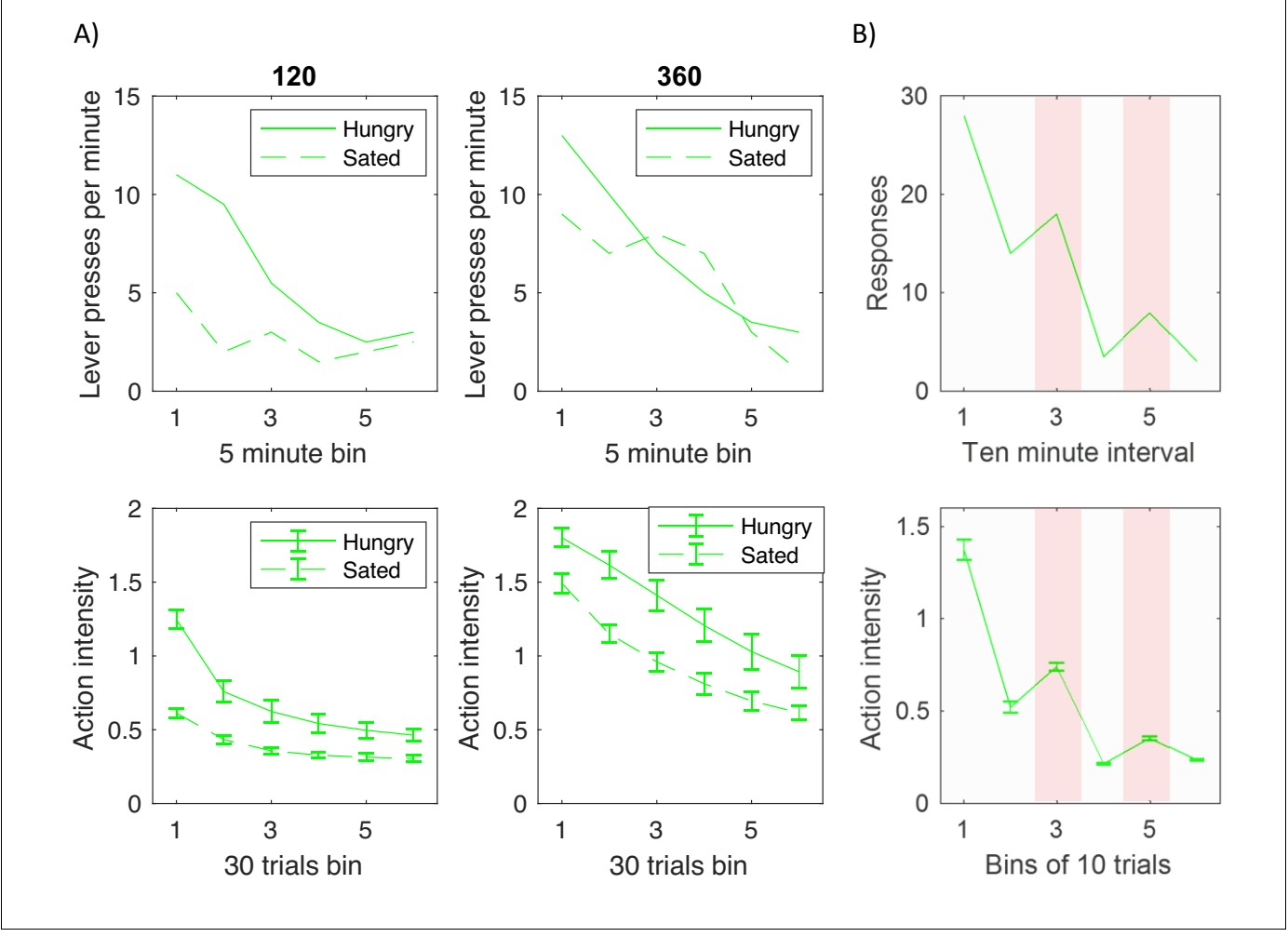

**Figure 8.** Comparison of experimentally observed lever pressing (top displays) and action intensity produced by the model (bottom displays). (A) Devaluation paradigm. Top displays replot data represented by open shapes in Figure 1 in a paper by *Dickinson et al., 1995*. Bottom displays show the average action intensity produced by the model across bins of 30 trials of the testing period. Simulations were repeated 10 times, and error bars indicate standard deviation across simulations. (B) Pavlovian-instrumental transfer. Top display replots the data represented by solid line in Figure 1 in a paper by *Estes, 1943*. Bottom displays show the average action intensity produced by the model across bins of 10 trials of the testing period. Simulations were repeated 10 times, and error bars indicate standard error across simulations.

in *Figure 7A*, because in these paradigms the reward per unit of time increases with frequency of lever press only to a certain point, but beyond certain frequency, there is no benefit of pressing faster.

To simulate the experiment described above, the model was trained either for 120 trials (bottom left display in *Figure 8A*) or 360 trials (bottom right display). During the training the reward depended on action as in *Figure 7A*. Following this training, the model was tested on 180 trials on which reward was not delivered, so in simulations $r = -a$ reflecting just a cost connected with making an effort. To simulate devaluation, the expectation of reward was set to 0.

Bottom displays in *Figure 8A* show the average action intensity produced by the model, and they reproduce qualitatively the key two effects in the top displays. First, the action intensity decreased with time, because the valuation and goal-directed systems learned that the reward was no longer available. Second, the action intensity just after devaluation was higher in the highly trained group (bottom right display) than in moderately trained group (bottom left display). This effect was produced by the model because after 360 trials of training the variance $\Sigma_h$ in the habit system was much lower than after 120 trials (right display in *Figure 7D*), so after the extended training, the action

intensity was to a larger extent determined by the habit system, which was not affected by devaluation.

The model can be easily extended to capture the phenomenon of Pavlovian-instrumental transfer. This phenomenon was observed in an experiment that consisted of three stages (*Estes, 1943*). First, animals were trained to press a lever to obtain a reward. Second, the animals were placed in a cage without levers, and trained that a conditioned stimulus predicted the reward. Third, the animals were placed back to a conditioning apparatus, but no reward was given for lever pressing. Top display in *Figure 8B* shows the numbers of responses in that third stage, and as expected they gradually decreased as animals learned that no reward was available. Importantly, in the third and fifth intervals of this testing phase the conditioned stimulus was shown (highlighted with pink background in *Figure 8B*), and then the lever pressing increased. Thus the learned association between the conditioned stimulus and reward influenced the intensity of actions produced in the presence of the stimulus.

The bottom display of *Figure 8B* shows the action intensity produced by the model in simulations of the above paradigm. As described in Materials and methods, the valuation system learned the rewards associated with two states: presence of a lever, and the conditioned stimulus. During the first stage (operant conditioning), the reward expectation computed by the valuation system drove action planning, while in the second stage (classical conditioning), no action was available, so the valuation system generated predictions for the reward without triggering action planning. In the third stage (testing), on the highlighted intervals on which the conditioned stimulus was present, the expected reward $v$ was increased, because it was a sum of rewards associated with both states. Consequently, the actor computed that a higher action intensity was required to obtain a bigger reward, because the goal-directed system assumes that the action intensity is proportional to the mean reward (orange Equation 3.2). In summary, the model explains the Pavlovian-instrumental transfer by proposing that the presence of the conditioned stimulus increases the reward expected by the valuation system, which results in actor selecting higher action intensity to obtain this anticipated reward.

## Extending the model to choice between two actions

This section shows how models developed within the DopAct framework can also describe more complex tasks with multiple actions and multiple dimensions of state. We consider a task involving choice between two options, often used in experimental studies, as it allows illustrating the generalization, and at the same time results in a relatively simple model. This section will also show that the models developed in the framework can under certain assumptions be closely related to previously proposed models of reinforcement learning and habit formation.

To make dimensionality of all variables and parameters explicit, we will denote vectors with a bar and matrices with a bold font. Thus $\bar{s}$ is a vector where different entries correspond to intensities of different stimuli in an environment, and $\bar{a}$ is a vector where different entries correspond to intensities of different actions. The model is set up such that only one action can be chosen, so following a decision, $a_i = 1$ for the chosen action $i$, while for other actions $a_{j \neq i} = 0$. Thus symbol $\bar{a}$ still denotes action intensity, but the intensity of an action only takes binary values once an action has been chosen.

Equation 9.1 in *Figure 9A* shows how the definitions of the probability distributions encoded by the goal-directed and habit systems can be generalized to multiple dimensions. Orange Equation 9.1 states that the reward expected by the goal-directed system has mean $\bar{a}^T \boldsymbol{Q} \bar{s}$, where $\boldsymbol{Q}$ is now a matrix of parameters. This notation highlights the link with the standard reinforcement learning, where the expected reward for selecting action $i$ in state $j$ is denoted by $Q_{i,j}$: Note that if $\bar{a}$ and $\bar{s}$ are both binary vectors with entries $i$ and $j$ equal to 1 in the corresponding vectors, and all other entries equal to 0, then $\bar{a}^T \boldsymbol{Q} \bar{s}$ is equal to the element $Q_{i,j}$ of matrix $\boldsymbol{Q}$.

In the model, the prior probability is proportional to a product of three distributions. The first of them is encoded by the habit system and given in blue Equation 9.1. The expected action intensity encoded in the habit system has mean $\boldsymbol{H}\bar{s}$, and this notation highlights the analogy with a recent model of habit formation (*Miller et al., 2019*) where a tendency to select action $i$ in state $j$ is also denoted by $H_{i,j}$. Additionally, we introduce another prior given in Equation 9.2, which ensures that

A)

Distributions: $P(R|\bar{a}, \bar{s}) = f(R; \bar{a}^T \boldsymbol{Q} \bar{s}, \Sigma_g) \quad P(\bar{a}|\bar{s}) = f(\bar{a}; \boldsymbol{H}\bar{s}, \Sigma_h)$ (9.1)

Additional priors: $\qquad P(a_1 a_2) = \exp(-a_1 a_2)$ (9.2)

$\qquad P(a_i) = 0 \;\; \text{for} \;\; a_i < 0 \;\; \text{or} \;\; a_i > 1$ (9.3)

B)

9.1-9.2 to 4.1:
$$F = \tfrac{1}{2}\left[-\ln \Sigma_g - \frac{(R - \bar{a}^T \boldsymbol{Q}\bar{s})^2}{\Sigma_g} - \ln \Sigma_h - \frac{(\bar{a} - \boldsymbol{H}\bar{s})^T(\bar{a} - \boldsymbol{H}\bar{s})}{\Sigma_h}\right] - a_1 a_2 \quad (9.4)$$

Planning:
$$\dot{\bar{a}} = \frac{\partial F}{\partial \bar{a}} = \delta_g \boldsymbol{Q}\bar{s} + (\boldsymbol{H}\bar{s} - \bar{a})/\Sigma_h - \begin{bmatrix} a_2 \\ a_1 \end{bmatrix} \quad (9.5)$$

Errors:
$$\delta_g = (R - \bar{a}^T \boldsymbol{Q}\bar{s})/\Sigma_g \qquad\qquad \bar{\delta}_h = \bar{a} - \boldsymbol{H}\bar{s} \quad (9.6)$$

Learning:
$$\Delta\boldsymbol{Q} \sim \frac{\partial F}{\partial \boldsymbol{Q}} = \delta_g \bar{a}\bar{s}^T \qquad\qquad \Delta\boldsymbol{H} \sim \frac{\partial F}{\partial \boldsymbol{H}} \sim \bar{\delta}_h \bar{s}^T \quad (9.7)$$

$$\Delta\Sigma_g \sim \frac{\partial F}{\partial \Sigma_g} \sim (\delta_g \Sigma_g)^2 - \Sigma_g \qquad \Delta\Sigma_h \sim \frac{\partial F}{\partial \Sigma_h} \sim \bar{\delta}_h^T \bar{\delta}_h - \Sigma_h \quad (9.8)$$

C)

Planning:
$$\dot{\bar{a}} \approx \underbrace{\frac{R}{\Sigma_g} \boldsymbol{Q}\bar{s}}_{\bar{I}_g} + \underbrace{\frac{1}{\Sigma_h} \boldsymbol{H}\bar{s}}_{\bar{I}_h} \quad (9.9)$$

Learning:
$$\Delta Q_{i,j} \sim r - Q_{i,j} \qquad \Delta H_{*,j} \sim a_* - H_{*,j} \quad (9.10)$$

**Figure 9.** Description of the model of choice between two actions. (**A**) Probability distributions assumed in the model. (**B**) Details of the algorithm. (**C**) Approximation of the algorithm. In blue Equation 9.10, * indicates that the parameters are updated for all actions.

only one action has intensity significantly deviating from 0. Furthermore, to link the framework with classical reinforcement learning, we enforce a third condition ensuring that action intensity remains between 0 and 1 (Equation 9.3). These additional priors will often result in one entry of $\bar{a}$ converging to 1, while all other entries decaying towards 0 due to competition. Since in our simulations we also use a binary state vector, the reward expected by the goal-directed system will often be equal to $Q_{i,j}$ as in the classical reinforcement learning (see paragraph above).

Let us now derive equations describing inference and learning for the above probabilistic model. Substituting probability densities from Equations 9.1 and 9.2 into the objective function of Equation 4.1, we obtain Equation 9.4 in *Figure 9B*. To ensure that action intensity remained between 0 and 1 (Equation 9.3), $a_i$ was set to one of these values if it exceeded the range during numerical integration.

To obtain the equations describing action planning or learning, we need to compute derivatives of $F$ over vectors or matrices. The rules for computing such derivatives are natural generalizations of the standard rules and they can be found in a tutorial paper (*Bogacz, 2017*). During planning, the action intensity should change proportionally to a gradient of $F$, which is given in Equation 9.5,

where the prediction errors are defined in Equations 9.6. These equations have an analogous form to those in *Figure 6A*, but are generalized to matrices. The only additional element is the last term in Equation 9.5, which ensures competition between different actions, i.e. $a_1$ will be decreased proportionally to $a_2$, and vice versa. During learning, the parameters need to be updated proportionally to the corresponding gradients of $F$, which are given in Equations 9.7 and 9.8. Again, these equations are fully analogous to those in *Figure 6A*.

Both action selection and learning in the above model share similarities with standard models of reinforcement learning and a recent model of habit formation (*Miller et al., 2019*). To see which action is most likely to be selected in the model, it is useful to consider the evolution of action intensity at the start of a trial, when $a_i \approx 0$, because the action with a largest initial input is likely to win the competition and be selected. Substituting orange Equation 9.6 into Equation 9.5 and setting $a_i = 0$, we obtain Equation 9.9 in *Figure 9C*. This equation suggests that probabilities of selecting actions depend on a sum of inputs form the goal-directed and habit systems weighted by their certainty, analogously as in a model by *Miller et al., 2019*. There are also similarities in the update rules: if only single elements of vectors $\bar{a}$ and $\bar{s}$ have non-zero values $a_i = 1$ and $s_j = 1$, then substituting Equations 9.6 into 9.7 and ignoring constants gives Equations 9.10. These equations suggest that the parameter $Q_{i,j}$ describing expected reward for action $i$ in state $j$ is modified proportionally to a reward prediction error, as in classical reinforcement learning. Additionally, for every action and current state $j$ the parameter describing a tendency to take this action is modified proportionally to a prediction error equal to a difference between the intensity of this action and the intensity expected by the habit system, as in a model of habit formation (*Miller et al., 2019*).

The similarity of a model developed in the DopAct framework to classical reinforcement learning, which has been designed to maximize resources, highlights that the model also tends to maximize resources, when animal's reserves are sufficiently low. But the framework is additionally adaptive to the levels of reserves: If the reserves were at the desired level, then $R = 0$ during action planning, so according to Equation 9.9, the goal-directed system would not suggest any action.

Let us now consider how the inference and learning can be implemented in a generalized version of the network described previously, which is shown in *Figure 10A*. In this network, striatum, output nuclei and thalamus include neural populations selective for the two alternative actions (shown in vivid and pale colours in *Figure 10A*), as in standard models of action selection in the basal ganglia (*Bogacz and Gurney, 2007*; *Frank et al., 2007*; *Gurney et al., 2001*). We assume that the connections between these nuclei are within the populations selective for a given action, as in previous models (*Bogacz and Gurney, 2007*; *Frank et al., 2007*; *Gurney et al., 2001*). Additionally, we assume that sensory cortex includes neurons selective for different states (shown in black and grey in

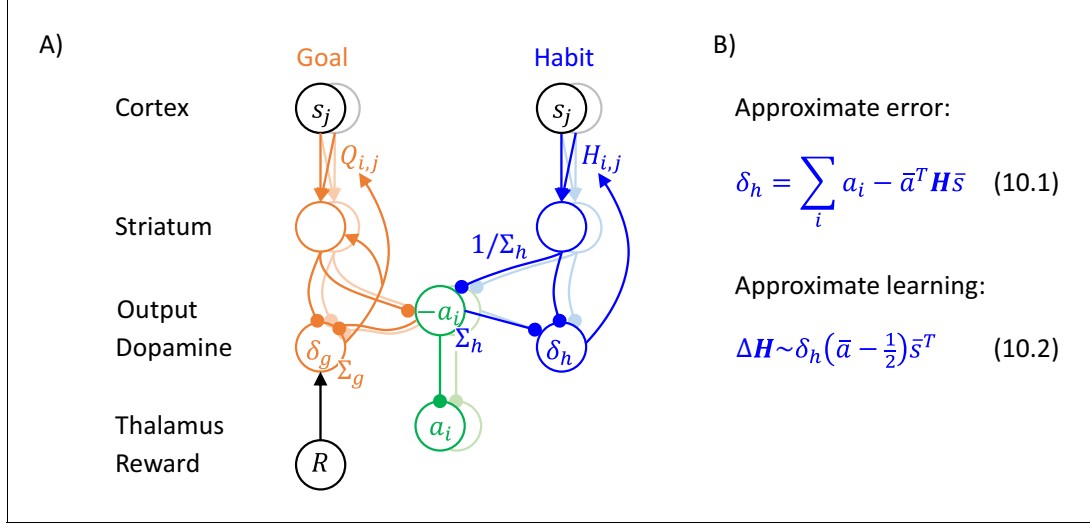

**Figure 10.** Implementation of the model of choice between two actions. (A) Mapping of the algorithm on network architecture. Notation as in *Figure 5B*. (B) An approximation of learning in the habit system.

*Figure 10A*), and the parameters $Q_{i,j}$ and $H_{i,j}$ are encoded in cortico-striatal connections. Then, the orange and blue terms in Equation 9.5 can be computed by the striatal neurons in goal-directed and habit systems in exactly analogous way as in the network inferring action intensity, and these terms can be integrated in the output nuclei and thalamus. The last term in Equation 9.5 corresponds to mutual inhibition between the populations selective for the two actions, and such inhibition could be provided by inhibitory projections that are presents in many different regions of this circuit, e.g. by co-lateral projections of striatal neurons (*Preston et al., 1980*) or via a subthalamic nucleus, which has been proposed to play role in inhibiting non-selected actions (*Bogacz and Gurney, 2007*; *Frank et al., 2007*; *Gurney et al., 2001*).

The prediction error in the goal-directed system (orange Equation 9.6) could be computed locally, because the orange dopaminergic neurons in *Figure 10A* receive inputs encoding all terms in the equation. During learning, the prediction error in the goal-directed system modulates plasticity of the corresponding cortico-striatal connections according to orange Equation 9.7, which describes a standard tri-factor Hebbian rule (if following movement the striatal neurons encode chosen action, as assumed in *Figure 5C*).

The prediction error in the habit system (blue Equation 9.6) is a vector, so computing it explicitly would also require multiple populations of dopaminergic neurons in the habit system selective for available actions, but different dopaminergic neurons in the real brain may not be selective for different actions (*da Silva et al., 2018*). Nevertheless, learning in the habit system can be approximated with a single dopaminergic population, because the prediction error $\bar{\delta}_h$ has a characteristic structure with large redundancy. Namely, if only one entry in the vectors $\bar{a}$ and $\bar{s}$ is equal to 1 and other entries to 0, then only one entry in $\bar{\delta}_h$ corresponding to the chosen action is positive, while all other entries are negative (because parameters $H_{i,j}$ stay in a range between 0 and 1 when initialized within this range and updated according to blue Equation 9.7). Hence, we simulated an approximate model just encoding the prediction error for the chosen action (Equation 10.1). With such a single modulatory signal, the learning rules for striatal neurons in the habit system have to be adjusted so the plasticity has opposite directions for the neurons selective for the chosen and the other actions. Such modified rule is given in Equation 10.2 and corresponds to tri-factor Hebbian learning (if striatal neurons in the habit system have activity proportional to $\bar{a}$ during learning, as we assumed for the goal-directed system). Thanks to this approximation, the prediction error and plasticity in the habit system take a form that is more analogous to that in the goal-directed system. When the prediction error in the habit system is a scalar, the learning rule for the variance parameter (blue Equation 9.8) becomes the same as in the model in the previous section (cf. blue Equation 6.5). Materials and method section provides the description of the valuation system in this model, and describes details of the simulations.

## Simulations of choice between two actions

To illustrate predictions made by the model, we simulated it in a probabilistic reversal task. On each trial, the model was 'presented' with one of two 'stimuli', that is one randomly chosen entry of vector $\bar{s}$ was set to 1, while the other entry was set to 0. On the initial 150 trials, the correct response was to select action 1 for stimulus 1 and action 2 for stimulus 2, while on the subsequent trials, the correct responses were reversed. The mean reward was equal to 1 for a correct response and 0 for an error. In each case, a Gaussian noise (with standard deviation $\sigma_r = 0.5$) was added to the reward.

*Figure 11A* shows changes in action intensity and inputs from goal-directed and habit systems as a function of time during planning on different trials within a simulation. On an early trial (left display) the changes in action intensity were primarily driven by the goal-directed system. The intensity of the correct action converged to 1, while it stayed at 0 for the incorrect one. After substantial training (middle display), the changes in action intensity were primarily driven by the faster habit system. Following a reversal (right display) one can observe a competition between the two systems: Although the goal-directed system had already learned the new contingency (solid orange curve), the habit system still provided larger input to the incorrect action node (dashed blue curve). Since the habit system was faster, the incorrect action had higher intensity initially, and only with time, the correct action node received input from the goal-directed system, and inhibited the incorrect one.

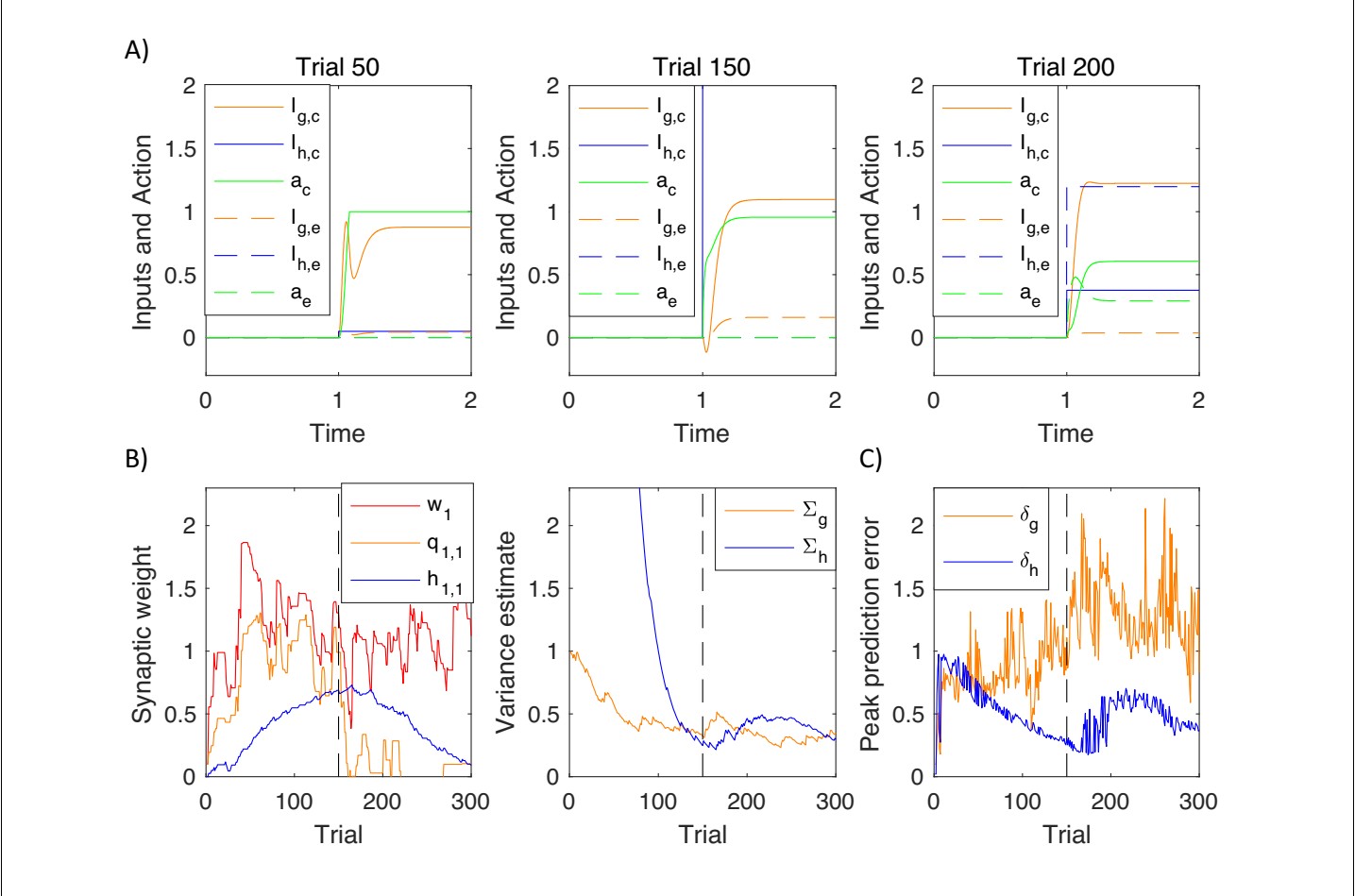

**Figure 11.** Simulation of the model of choice between two actions. (**A**) Changes in action intensity and inputs from the goal-directed and habit systems, defined below Equation 9.9. Solid lines correspond to a correct action and dashed lines to an error. Thus selecting action 1 for stimulus 1 (or action 2 for stimulus 2) corresponds to solid lines before reversal (left and middle displays) and to dashed lines after reversal (right display). (**B**) Changes in model parameters across trials. Dashed black lines indicate a reversal trial. (**C**) Maximum values of prediction errors during action planning on each simulated trial.

*Figure 11B* shows how parameters in the model changed over trials. Left display illustrates changes in sample cortico-striatal weights in the three systems. The valuation system rapidly learned the reward available, but after reversal this estimate decreased, as the model persevered in choosing the incorrect option. Once the model discovered the new rule, the estimated value of the stimulus increased. The goal-directed system learned that selecting the first action after the first stimulus gave higher rewards before reversal, but not after. The changes in the parameters of the habit system followed those in the goal-directed system. The right display shows that the variance estimated by the habit system initially decreased, but then increased several trials after the reversal, when the goal-directed system discovered the new contingency, and thus the selected actions differed from the habitual ones. *Figure 11C* shows an analogous pattern in dopaminergic activity, where the neurons in the habit system signalled higher prediction errors following a reversal. This pattern of prediction errors is unique to the habit system, as the prediction errors in the goal-directed system (orange curve) fluctuated throughout the simulation following the fluctuations in reward. The increase in dopaminergic activity in the habit system following a reversal is a key experimental prediction of the model, to which we will come back in Discussion.

Let us consider the mechanisms of reversal in the model. Since the prediction errors in the habit system do not directly depend on rewards, the habit system would not perform reversal on its own, and the goal-directed system is necessary to initiate the reversal. This feature is visible in simulations,

where just after the reversal the agent was still selecting the same actions as before, so the habits were still being strengthen rather weakened (the blue curve in left display of *Figure 11B* still increased for ~20 trials after the reversal). When the goal-directed system learned that the previously selected actions were no longer rewarded, the tendency to select them decreased, and other actions had higher chances of being selected due to noise (although the amount of noise added to the choice process was constant, there was a higher chance for noise to affect behaviour, because the old actions were now suggested only by the habit rather than both systems). Once the goal-directed system found that the actions selected according to new contingency gave rewards, the probability of selecting action according to the old contingency decreased, and only then the habit system slowly unlearned the old habit.

It is worth adding that the reversal was made harder by the fact that a sudden change in reward increased the uncertainty of the goal-directed system (the orange curve in the right display of *Figure 11B* increased after reversal), which actually weakened the control by that system. Nevertheless, this increase of uncertainty was brief, because the goal-directed system quickly learned to predict rewards in the new contingency and regained its influence on choices.

## Discussion

In this paper, we proposed how an action can be identified through Bayesian inference, where the habit system provides a prior and the goal-directed system represents reward likelihood. Within the DopAct framework, the goal-directed and habit systems may not be viewed as fundamentally different systems, but rather as analogous segments of neural machinery performing inference in a hierarchical probabilistic model (*Figure 1A*), which correspond to different levels of hierarchy.

In this section, we discuss the relationship of the framework to other theories and experimental data, mechanisms of habit formation, and suggest experimental predictions and directions for future work.

### Relationship to other theories

The DopAct framework combines elements from four theories: reinforcement learning, active inference, habit formation, and planning as inference. For each of the theories we summarize key similarities, and highlight the ways in which the DopAct framework extends them.

As in classical reinforcement learning (*Houk et al., 1995*; *Montague et al., 1996*), in the DopAct framework the dopaminergic neurons in the valuation and goal-directed systems encode reward prediction errors, and these prediction errors drive learning to improve future choices. However, the key conceptual difference of the DopAct framework is that it assumes that animals aim to achieve a desired level of reserves (*Buckley et al., 2017*; *Hull, 1952*; *Stephan et al., 2016*), rather than always maximize acquiring resources. It has been proposed that when a physiological state is considered, the reward an animal aims to maximize can be defined as a reduction of distance between the current and desired levels of reserves (*Juechems and Summerfield, 2019*; *Keramati and Gutkin, 2014*). Under this definition, a resource is equal to such subjective reward only if consuming it would not bring the animal beyond its optimal reserve level. When an animal is close to the desired level, acquiring a resource may even move the animal further from the desired level, resulting in a negative subjective reward. As the standard reinforcement learning algorithms do not consider physiological state, they do not always maximize the subjective reward defined in this way. By contrast, the DopAct framework offers flexibility to stop acquiring resources, when the reserves reach the desired level.

The DopAct framework relies on a key high-level principle from the active inference theory (*Friston, 2010*) that the prediction errors can be minimized by both learning and action planning. Furthermore, the network implementations of the proposed models share a similarity with predictive coding networks that the neurons encoding prediction errors affect both the plasticity and the activity of its target neurons (*Friston, 2005*; *Rao and Ballard, 1999*). A novel contribution of this paper is to show how these principles can be realized in anatomically identified networks in the brain.

The DopAct framework shares a feature of a recent model of habit formation (*Miller et al., 2019*) that learning in the habit system is driven by prediction errors that do not depend on reward, but rather encode the difference between the chosen and habitual actions. The key new contribution of

this paper is to propose how such learning can be implemented in the basal ganglia circuit including multiple populations of dopaminergic neurons encoding different prediction errors.

Similarly as in the model describing goal-directed decision making as probabilistic inference (*Solway and Botvinick, 2012*), the actions selected in the DopAct framework maximize a posterior probability of action given the reward. The new contribution of this paper is making explicit the rationale for why such probabilistic inference is the right thing for the brain to do: The resource that should be acquired in a given state depends on the level of reserves, so the inferred action should depend on the reward required to restore the reserves. We also proposed a detailed implementation of the probabilistic inference in the basal ganglia circuit.

It is useful to discuss the relationship of the DopAct framework to several other theories. The tonic level of dopamine has been proposed to determine the vigour of movements (*Niv et al., 2007*). In our model selecting action intensity, the dopaminergic signals in the valuation and goal-directed systems indeed influence the resulting intensity of movement, but in the DopAct framework, it is the phasic rather than tonic dopamine that determines the vigour, in agreement with recent data (*da Silva et al., 2018*). It has been also proposed that dopamine encodes incentive salience of the available rewards (*Berridge and Robinson, 1998*; *McClure et al., 2003*). Such encoding is present in the DopAct framework, where the prediction error in the goal-directed system depends on whether the available resource is desired by an animal.

## Relationship to experimental data

To relate the DopAct framework to experimental data, we need to assume a particular mapping of different systems on anatomically defined brain regions. Thus we assume that the striatal neurons in valuation, goal-directed, and habit systems can be approximately mapped on ventral, dorsomedial, and dorsolateral striatum. This mapping is consistent with the pattern of neural activity in the striatum, which shifts from encoding reward expectation to movement as one progresses from ventral to dorsolateral striatum (*Burton et al., 2015*), and with increased activity in dorsolateral striatum during habitual movements (*Tricomi et al., 2009*). This mapping is also consistent with the observation that deactivation of dorsomedial striatum impairs learning which action leads to larger rewards (*Yin et al., 2005*), while lesion of dorsolateral striatum prevents habit formation (*Yin et al., 2004*). Furthermore, we will assume that dopaminergic neurons in valuation, goal-directed, and habit systems can be mapped on a spectrum of dopaminergic neurons ranging from ventral tegmental area (VTA) to substantia nigra pars compacta (SNc). VTA is connected with striatal regions we mapped on the valuation system, while SNc with those mapped on the habit system (*Haber et al., 2000*), so we assume that $\delta_v$ and $\delta_h$ are represented in VTA and SNc respectively. Such mapping in consistent with lesions to SNc preventing habit formation (*Faure et al., 2005*). The mapping of the dopaminergic neurons from the goal-directed system is less clear, so let us assume that these neurons may be present in both areas.

The key prediction of the DopAct framework is that the dopaminergic neurons in the valuation and goal-directed systems should encode reward prediction errors, while the dopaminergic neurons in the habit system should respond to non-habitual actions. This prediction can be most directly compared with the data in a study where rewards and movements have been dissociated. That study employed a task in which mice could make spontaneous movements and rewards were delivered at random times (*Howe and Dombeck, 2016*). It has been observed that a fraction of dopaminergic neurons had increased responses to rewards, while a group of neurons responded to movements. Moreover, the reward responding neurons were located in VTA while most movement responding neurons in SNc (*Howe and Dombeck, 2016*). In that study the rewards were delivered to animals irrespectively of movements, so the movements they generated were most likely not driven by processes aiming at achieving reward (simulated in this paper), but rather by other inputs (modelled by noise in our simulations). To relate this task to the DopAct framework, let us consider the prediction errors likely to occur at the times of reward and movement. At the time of reward the animal was not able to predict it, so $\delta_v > 0$, $\delta_g > 0$, but it was not necessarily making any movements $\delta_h = 0$, while at the time of a movement the animal might have not expected reward $\delta_v = \delta_g = 0$, but might have made non-habitual movements $\delta_h > 0$. Hence the framework predicts separate groups of dopaminergic neurons to produce responses at times of reward and movements, as experimentally observed (*Howe and Dombeck, 2016*). Furthermore, the peak of the movement related response of SNc neurons was observed to occur after the movement onset (*Howe and Dombeck, 2016*), which suggests

that most of this dopaminergic activity was a response to a movement rather than activity initiating a movement. This timing is consistent with the role of dopaminergic neurons in the habit system, which compute a movement prediction error, rather than initiate movements.

While discussing dopaminergic neurons, one has to mention the influential studies showing that VTA neurons encode reward prediction error (*Eshel et al., 2016*; *Schultz et al., 1997*; *Tobler et al., 2005*). So for completeness, let us reiterate that in the DopAct framework the valuation system is similar to the standard temporal difference learning model, hence it inherits the ability to account for the dopaminergic responses to unexpected rewards previously explained with that model (*Figure 7B*).

The DopAct framework also makes predictions on dopaminergic responses during movements performed to obtain rewards. In presented simulations, such responses were present in all systems (*Figure 7B–C*), and indeed responses to reward-directed movements were observed experimentally in both VTA and SNc (*Engelhard et al., 2019*; *Schultz, 1986*). The framework predicts that the responses to movements should be modulated by the magnitude of available reward in the valuation and goal-directed systems, but not in the habit system. This prediction can be compared with data from a task in which animals could press one of two levers that differed in magnitude of resulting rewards (*Jin and Costa, 2010*). So for this task, the framework predicts that the dopaminergic neurons in the valuation and goal-directed systems should respond differently depending on which lever was pressed, while the dopaminergic response in the habit system should depend just on action intensity but not reward magnitude. Indeed, a diversity of dopaminergic neurons have been observed in SNc, and the neurons differed in whether their movement related response depended on reward available (Figure 4j in the paper by *Jin and Costa, 2010*).

In the DopAct framework, the activity of dopaminergic neurons in the goal-directed system is normalized by the uncertainty of that system. Analogous scaling of dopaminergic activity by an estimate of reward variance is also present in a model by *Gershman, 2017*. He demonstrated that such scaling is consistent with an experimental observation that dopaminergic responses adapt to the range of rewards available in a given context (*Tobler et al., 2005*).

In the DopAct framework the role of dopamine during action planning is specific to preparing goal-directed but not habitual movements (*Figure 7E*). Thus the framework is consistent with an observation that blocking dopaminergic transmission slows responses to reward-predicting cues early in training, but not after extensive training, when the responses presumably became habitual (*Choi et al., 2005*). Analogously, the DopAct framework is consistent with an impairment in Parkinson's disease for goal-directed but not habitual choices (*de Wit et al., 2011*) or voluntary but not cue driven movements (*Johnson et al., 2016*). The difficulty in movement initiation in Parkinson's disease seems to depend on whether the action is voluntary or in response to a stimulus, so even highly practiced movements like walking may be difficult if performed voluntarily, but easier in response to auditory or visual cues (*Rochester et al., 2005*). Such movements performed to cues are likely to engage the habit system, because responding to stimuli is a hallmark of habitual behaviour (*Dickinson and Balleine, 2002*).

Finally, let us discuss a feature of the DopAct framework related to the dynamics of competition between systems during action planning. Such competition is illustrated in the right display of *Figure 11A*, where after a reversal, the faster habit system initially prepared an incorrect action, but later the slower goal-directed system increased the intensity of the correct action. Analogous behaviour has been shown in a recent study, where human participants were extensively trained to make particular responses to given stimuli (*Hardwick et al., 2019*). After a reversal, they tended to produce incorrect habitual actions when required to respond rapidly, but were able to produce the correct actions given sufficient time.

## Mechanisms of habitual behaviour

Since the mechanisms of habit formation in the DopAct framework fundamentally differ from a theory widely accepted by a computational neuroscience community (*Daw et al., 2005*), this section is dedicated to comparing the two accounts, and discussing the properties of the habit system in the framework.

An influential theory suggests that two anatomically separate systems in the brain underlie goal-directed and habitual behaviour and a competition between them is resolved according to uncertainty of the systems (*Daw et al., 2005*). The DopAct framework agrees with these general principles

but differs from the theory of *Daw et al., 2005* in the nature of computations in these systems, and their mapping on brain anatomy. *Daw et al., 2005* proposed that goal-directed behaviour is controlled by a cortical model-based system that learns the transitions between states resulting from actions, while habitual behaviour arises from a striatal model-free system that learns policy according to standard reinforcement learning. By contrast, the DopAct framework suggests that goal-directed behaviour in simple lever-pressing experiments does not require learning state transitions, but such behaviour can be also supported by a striatal goal-directed system that learns expected rewards from actions in a way similar to standard reinforcement learning models. So in the DopAct framework it is the goal-directed rather than habit system that learns according to reward prediction error encoded by dopaminergic neurons. Furthermore, in the DopAct framework (following the model by *Miller et al., 2019*) habits arise simply from repeating actions, so their acquisition is not directly driven by reward prediction error, unlike in the model of *Daw et al., 2005*.

The accounts of habit formation in the DopAct framework and the model of *Daw et al., 2005* make different predictions. Since the theory of *Daw et al., 2005* assumes that a system underlying habitual behaviour learns with standard reinforcement learning, it predicts that striatal neurons supporting habitual behaviour should receive reward prediction error. However, the dopaminergic neurons that have been famously shown to encode reward prediction error (*Schultz et al., 1997*) are located in VTA, which does not send major projections to the dorsolateral striatum underlying habitual behaviour. These striatal neurons receive dopaminergic input from SNc (*Haber et al., 2000*), and it is questionable to what extent dopaminergic neurons in SNc encode reward prediction error. Although such encoding has been reported (*Zaghloul et al., 2009*), studies which directly compared the activity of VTA and SNc neurons demonstrated that neurons encoding reward prediction error are significantly more frequent in VTA than SNc (*Howe and Dombeck, 2016*; *Matsumoto and Hikosaka, 2009*). So the striatal neurons underlying habitual behaviour do not seem to receive much of the teaching signal that would be expected if habit formation arose from the processes of reinforcement learning proposed by *Daw et al., 2005*. By contrast, the DopAct framework assumes that the habit system learns on the basis of a teaching signal encoding how the chosen action differs from the habitual one, so it predicts that SNc neurons should respond to non-habitual movements. It has indeed been observed that the dopaminergic neurons in SNc respond to movements (*Howe and Dombeck, 2016*; *Schultz et al., 1983*), but it has not been systematically analysed yet if these responses preferentially encode non-habitual movements (we will come back to this key prediction in the next section).

It is worth discussing how the habits may be suppressed if previously learnt habitual behaviour is no longer appropriate. In the DopAct framework, old habits die hard. When the habitual behaviour is no longer rewarded, the negative reward prediction errors do not directly suppress the behaviour in the habit system. So, as mentioned at the end of the Results section, in order to reverse behaviour, the control cannot be completely taken over by the habit system, but the goal-directed system needs to provide at least some contribution to action planning to initiate the reversal when needed. Nevertheless, simulations presented in this paper show that for certain parameters the control of habit system may be released when no longer required, and the model can reproduce the patterns of behaviour observed in extinction experiments (*Figure 8*). However, simulations by *Miller et al., 2019* show that their closely related model can sometimes persist in habitual behaviour even if it is not desired. Therefore, it is possible that there may exist other mechanisms that may help the goal-directed system to regain control if habitual behaviour ceases to be appropriate. For example, it has been proposed that a sudden increase in prediction errors occurring when environment changes may attract attention and result in the goal-directed system taking charge of animals' choices (*FitzGerald et al., 2014*).

Finally, let us discuss the relationship of the DopAct framework to an observation that habits are more difficult to produce in variable ratio schedules than variable interval schedules (*Dickinson et al., 1983*). In the variable ratio schedules a lever press is followed by a reward with a fixed probability $p$. By contrast in the variable interval schedule a lever press is followed by a reward only if the reward is 'available'. Just after consuming a reward, lever pressing has no effect, and another reward may become "available" as time goes on with a fixed probability per unit of time. An elegant explanation for why habit formation depends on the schedule has been provided by *Miller et al., 2019*, and a partially similar explanation can be given within the DopAct framework, as we now summarize. *Miller et al., 2019* noticed that reward rate as a function of action frequency

follows qualitatively different relationships in different schedules. In particular, in the variable ratio schedule the expected number of rewards per unit time is directly proportional to number of lever presses, i.e. $E(r) = pa$. By contrast, in the variable interval schedule, the reward rate initially increases with the number of level presses, but beyond some frequency there is little benefit of responding more often, so the reward rate is a nonlinear saturating function of action frequency. The model selecting action intensity in the DopAct framework assumes a linear dependence of mean reward on action intensity (orange Equation 3.2), so in the variable ratio schedule, it will learn $q = p$, and then predict mean reward accurately no matter what action intensity is selected. By contrast, in the variable interval schedule the predictions will be less accurate, because the form of the actual dependence of reward on action frequency is different to that assumed by the model. Consequently, the reward uncertainty of the goal-directed system $\Sigma_g$ is likely to be lower in the variable ratio than variable interval schedule. This decreased uncertainty makes the goal-directed system less likely to give in to the habit system, resulting in less habitual behaviour in the variable ratio schedule.

## Experimental predictions

We start with describing two most critical predictions of the DopAct framework, testing of which may validate or falsify the two key assumptions of the framework, and next we discuss other predictions. The first key prediction of the DopAct framework is that the dopaminergic neurons in the habit system should respond to movements more, when they are not habitual, e.g. at an initial phase of task acquisition or after a reversal (*Figure 11C*). This prediction could be tested by monitoring the activity of dopaminergic neurons projecting to dorsolateral striatum in a task where animals are trained to perform a particular response for sufficiently long that it becomes habitual, and then the required response is reversed. The framework predicts that these dopaminergic neurons should have higher activity during initial training and in a period after the reversal, than during the period when the action is habitual.

The second key prediction follows from a central feature of the DopAct framework that the expectation of the reward in the goal-directed system arises from forming a motor plan to obtain it. Thus the framework predicts that the dopaminergic responses in the goal-directed system to stimuli predicting a reward should last longer if planning actions to obtain the reward takes more time, or if an animal is prevented from making a response. One way to test this prediction would be to optogenetically block striatal neurons expressing D1 receptors in the goal-directed system for a fixed period after the onset of a stimulus, so the action plan cannot be formed. The framework predicts that such manipulation should prolong the response of dopaminergic neurons in that system. Another way of testing this prediction would be to employ a task where goal-directed planning becomes more efficient and thus shorter with practice. The framework predicts that in such tasks the responses of dopaminergic neurons in the goal-directed system during action planning should get briefer with practice, and their duration should be correlated with reaction time across stages of task acquisition.

The DopAct framework also predicts distinct patterns of activity for different populations of dopaminergic neurons. As already mentioned above, dopaminergic neurons in the habit system should respond to movements more, when they are not habitual. When the movements become highly habitual, these neurons should tend to more often produce brief decreases in response (*Figure 7C*, right). Furthermore, when the choices become mostly driven by the habit system, then dopaminergic neurons in the goal-directed system should no longer signal reward prediction error after stimulus (*Figure 7C*, right). By contrast, the dopaminergic neurons in the valuation system should signal reward prediction error after stimulus even once the action becomes habitual (*Figure 7B*).

Patterns of prediction errors expected from the DopAct framework could also be investigated with fMRI. Models developed within the framework could be fitted to behaviour of human participants performing choice tasks. Such models could then generate patterns of different prediction errors ($\delta_v$, $\delta_g$, $\delta_h$) expected on individual trials. Since prediction errors encoded by dopaminergic neurons are also correlated with striatal BOLD signal (*O'Doherty et al., 2004*), one could investigate if different prediction errors in the DopAct framework are correlated with BOLD signal in different striatal regions.

In the DopAct framework dopaminergic neurons increase the gain of striatal neurons during action planning, only in the goal-directed but not in the habit system. Therefore, the framework

predicts that the dopamine concentration should have a larger effect on the slope of firing-Input curves for the striatal neurons in the goal-directed than the habit system. This prediction may seem surprising, because striatal neurons express dopaminergic receptors throughout the striatum (*Huntley et al., 1992*). Nevertheless, it is consistent with reduced effects of dopamine blockade on habitual movements (*Choi et al., 2005*) that are known to rely on dorsolateral striatum (*Yin et al., 2004*). Accordingly, the DopAct framework predicts that the dopaminergic modulation in dorsolateral striatum should primarily affect plasticity rather than excitability of neurons.

## Directions for future work

This paper described a general framework for understanding the function of dopaminergic neurons in the basal ganglia, and presented simple models capturing only a subset of experimental data. To describe responses observed in more realistically complex tasks, models could be developed following a similar procedure as in this paper. Namely, a probabilistic model could be formulated for a task, and a network minimizing the corresponding free-energy derived, simulated and compared with experimental data. This section highlights key experimental observations the models described in this paper are unable to capture, and suggests directions for developing models consistent with them.

The presented models do not mechanistically explain the dependence of dopamine release in ventral striatum on motivational state such as hunger or thirst (*Papageorgiou et al., 2016*). To reproduce these activity patterns, it will be important to extend the framework to describe the computations in the valuation system. It will also be important to better understand the interactions between the valuation and goal-directed systems during the choice of action intensity. In the presented model, the selected action intensity depends on the value of the state estimated by the valuation system, and conversely, the produced action intensity influences reward and thus the value learned by the valuation system. In the presented simulations the parameters (e.g. learning rates) were chosen such that the model learned to select action intensity giving highest reward, but such behaviour was not present for all parameter values. Hence it needs to be understood how the interactions between the valuation and goal-directed systems need to be set up so the model robustly finds the action intensity giving the maximum reward.

The models do not describe how the striatal neurons distinguish whether dopaminergic prediction error should affect their plasticity or excitability, and for simplicity, in the presented simulations we allowed the weights to be modified only when reward was presented. However, the same dopaminergic signal after a stimulus predicting reward may need to trigger plasticity in one group of striatal neurons (selective for a past action that led to this valuable state), and changes in excitability in another group (selective for a future action). It will be important to further understand the mechanisms which can be employed by striatal neurons to appropriately react to dopamine signals (*Berke, 2018*; *Mohebi et al., 2019*).

The models presented in this paper described only a part of the basal ganglia circuit, and it will be important to include also other elements of the circuit. In particular, this paper focussed on a subset of striatal neurons expressing D1 receptors, which project directly to the output nuclei and facilitate movements, but another population expressing D2 receptors projects via an indirect pathway and inhibits movements (*Kravitz et al., 2010*). Computational models suggest that these neurons predominantly learn from negative feedback (*Collins and Frank, 2014*; *Mikhael and Bogacz, 2016*; *Möller and Bogacz, 2019*) and it would be interesting include their role in preventing unsuitable movements in the DopAct framework.

The basal ganglia circuit also includes a hyperdirect pathway, which contains the subthalamic nucleus. It has been proposed that a function of the subthalamic nucleus is to inhibit non-selected actions (*Gurney et al., 2001*), and the hyperdirect pathway may support the competition between actions that is present in the framework. The subthalamic nucleus has also been proposed to be involved in determining when the planning process should finish and action should be initiated (*Frank et al., 2007*). For simplicity, in this paper the process of action planning has been simulated for a fixed interval (until time $t = 2$ in *Figures 7* and *11*). It will be important to extend the framework to describe the mechanisms initiating an action. If actions were executed as soon as a motor plan is formed, the increase in the habit prediction error would be briefer than that depicted in *Figure 7C*. In such an extended model the valuation and goal-directed systems would also need to be modified to learn to expect reward at a particular time after the action.

The presented models cannot reproduce the ramping of dopaminergic activity, observed as animals approached rewards (*Howe et al., 2013*). To capture these data, the valuation system could incorporate synaptic decay that has been shown to allow standard reinforcement learning models to reproduce the ramping of prediction error (*Kato and Morita, 2016*).

It has been also observed that dopaminergic neurons respond not only to unexpected magnitude of reward, but also when the type of reward differs from that expected (*Takahashi et al., 2017*). To capture such prediction errors, the framework could be extended to assume that each system tries to predict multiple dimensions of reward or movement (cf *Gardner et al., 2018*).

Finally, dopaminergic neurons also project to regions beyond basal ganglia, such as amygdala, which plays a role in habit formation (*Balleine et al., 2003*), and cortex, where they have been proposed to modulate synaptic plasticity (*Roelfsema and van Ooyen, 2005*). It would be interesting to extend the DopAct framework to capture dopamine role in learning and action planning in these regions.

## Materials and methods

This section describes details of simulations of models developed within the DopAct framework for two tasks: selecting action intensity and choice between two actions. The models were simulated in Matlab (RRID:SCR_001622), and all codes are available at MRC Brain Network Dynamics Unit Data Sharing Platform (https://data.mrc.ox.ac.uk/data-set/simulations-action-inference).

### Selecting action intensity

We first describe the valuation system, and then provide details of the model in various simulated scenarios.

The valuation system was based on the standard temporal difference model (*Montague et al., 1996*). Following that model we assume that the valuation system can access information on how long ago a stimulus was presented. In particular, we assume that time can be divided into brief intervals of length $I$. The state of the environment is represented by a column vector $\bar{s}_v$ with entries corresponding to individual intervals, such that $s_{v,1} = 1$ if the stimulus has been present in the current interval, $s_{v,2} = 1$ if the stimulus was present in the previous interval, etc. Although more realistic generalizations of this representation have been proposed (*Daw et al., 2006*; *Ludvig et al., 2008*), we use this standard representation for simplicity.

*Figure 12A* lists equations describing the valuation system, which are based on temporal difference learning but adapted to continuous time. According to Equation 12.1, the estimate of the value of state $s$ converges in equilibrium to $v = \bar{w}\bar{s}_v$, where $\bar{w}$ denotes a row vector of parameters

A)

$$\tau\dot{v} = \bar{w}\bar{s}_v - v \qquad (12.1)$$

$$\tau_\delta\dot{\delta}_v = r + v - v_{t-I} - \delta_v \qquad (12.2)$$

$$\dot{\bar{w}} = \alpha_v\,\delta_v\,\bar{e} \qquad (12.3)$$

$$\tau\dot{\bar{e}} = \lambda\bar{e}_{t-I-3\tau} + \bar{s}_v^T - \bar{e} \qquad (12.4)$$

$$\tau\dot{r} = r_0 - r \qquad (12.5)$$

B)

$$\tau\dot{v} = \bar{w}\bar{s} - v \qquad (12.6)$$

$$\Delta\bar{w} = \alpha_v(r - v)\bar{s} \qquad (12.7)$$

**Figure 12.** Description of the valuation system. (**A**) Temporal difference learning model used in simulations of action intensity selection. (**B**) A simplified model used in simulations of choice between two actions.

describing how much reward can be expected after stimulus appearing in a particular interval. Equation 12.2 describes the dynamics of the prediction error in the valuation system, which converges to a difference between total reward ($r + v$) and the expectation of that reward made at a previous interval ($v_{t-1}$), as in the standard temporal difference learning (**Sutton and Barto, 1998**). The weight parameters are modified proportionally to the prediction error as described by Equation 12.3, where $\alpha_v$ is a learning rate, and $\bar{e}$ are eligibility traces associated with weights $\bar{w}$, which describe when the weights can be modified. In basic reinforcement learning $\bar{e} = \bar{s}_v^T$, i.e. a weight can only be modified if the corresponding state is present. Equation 12.4 describes the dynamics of the eligibility traces, and if one ignored the first term on the right, it would converge to $\bar{e} = \bar{s}_v^T$. The first term on the right of Equation 12.4 ensures that the eligibility traces persist over time, and parameter $\lambda$ describes what fraction of the eligibility traces survives from one interval to the next (**Ludvig et al., 2008**). Such persistent eligibility traces are known to speed up learning (**Sutton and Barto, 1998**). The first term on the right of Equation 12.4 includes an eligibility trace from time $t - I - 3\tau$, that is from a time slightly further than one interval ago, to avoid the influence of transient dynamics occurring at the transition between intervals. It is also ensured in the simulations that parameters $\bar{w}$ do not become negative, as the desired reward value $v$ computed by the valuation system should not be negative. Thus if any element of $\bar{w}$ becomes negative, it is set to 0. Finally, Equation 12.5 describes the dynamics of the reward signal $r$, which follows the actual value to reward $r_0$. This dynamics has been introduced so that the reward signal rises with the same rate as the value estimate (the same time constant is used in Equations 12.1 and 12.5), and these quantities can be subtracted to result in no prediction error when the reward obtained is equal to that predicted by the valuation system.

In simulations involving selection of action intensity, the time represented by the valuation system was divided into intervals of $I = 0.2$. The stimulus was presented at time $t = 1$, while the reward was given at time $t = 2$, thus the valuation system represented the value of 5 time intervals (i.e. vectors $\bar{w}$, $\bar{s}_v$ and $\bar{e}$ had 5 elements each). The parameters controlling retention of eligibility trace was set to $\lambda = 0.9$. The state provided to the actor was equal to $s = 1$ from time $t = 1$ onwards. We assumed that the intensity of action executed by the agent was equal to the inferred action intensity plus motor noise with standard deviation $\sigma_a = 1$ (this random number was added to action intensity at time $t = 2$). During intervals in which rewards were provided (from $t = 2$ onwards) the parameters were continuously updated according to Equations 6.8-9. In simulations the learning rates were set to: $\alpha_v = 0.5$, $\alpha_g = 0.05$, $\alpha_h = 0.02$, $\alpha_{\Sigma g} = 0.05$, $\alpha_{\Sigma h} = 0.1$. The time constants were set to: $\tau = 0.05$, $\tau_\delta = 0.02$, and the differential equations were solved numerically using Euler method with integration step 0.001. The model parameters were initialized to: $v_i = q = 0.1$, $h = 0$, $\Sigma_g = 1$ and $\Sigma_h = 100$.

To simulate devaluation, the expectation of reward was set to 0 by setting $v_i = q = 0$, as a recent modelling study suggests that such scaling of learned parameters by motivational state is required for reproducing experimentally observed effects of motivational state on dopaminergic responses encoding reward prediction error (**van Swieten and Bogacz, 2020**).

In the simulations of Pavlovian-instrumental transfer, the valuation system was learning the values of two states corresponding to the presence of the lever and the conditioned stimulus. Thus the state vector $\bar{s}_v$ had 10 entries, where the first 5 entries were set to 1 at different intervals after 'lever appearance', while the other 5 entries were set to 1 at different intervals after conditioned stimulus. Consequently, the vector of parameters of the valuation system $\bar{w}$ also had 10 entries. The simulations of the first stage (operant conditioning) consisted of 100 trials in which the model was trained analogously as in the simulations described in the above paragraph. At this stage only first 5 entries of vector $\bar{s}_v$ could take non-zero values, and hence only the first 5 entries of $\bar{w}$ were modified. The state provided to the actor was equal to $s = 1$ when 'lever appeared' that is from time $t = 1$ onwards. The simulations of the second stage (classical conditioning) consisted of 100 trials in which only the valuation system was learning. At this stage, the conditioned stimulus was presented at time $t = 1$, and the reward $r = 1$ was given at time $t = 2$, thus $s_{v,6} = 1$ for $t \in [1, 1.2]$, $s_{v,7} = 1$ for $t \in [1.2, 1.4]$, etc. The simulations of the third stage (testing) consisted of 60 trials in which only negative reward accounting for effort $r = -a$ was given. On trials 21-30 and 41-50, both 'lever and conditioned stimulus were presented', that is $s_{v,1} = s_{v,6} = 1$ for $t \in [1, 1.2]$, etc., while on the other trials only the 'lever was presented'. The model was simulated with the same parameters as described in the previous

paragraph, except for modified values of two learning rates $\alpha_g = 0.015$, $\alpha_h = 0.005$, to reproduce the dynamics of learning shown by experimental animals.

In all simulations in this paper, a constraint (or a 'hyperprior') on the minimum value of the variance parameters was introduced, such that if $\Sigma_g$ or $\Sigma_h$ decreased below 0.2, it was set to 0.2.

### Choice between two actions

Analogously, as in the previous section, we first describe the valuation system, and then provide the details of the simulations.

In the simulations of choice, we used a simplified version of the valuation system, which for each state $j$ learns a single parameter $w_j$ (rather than the vector of parameters encoding the reward predicted in different moments in time). The equations describing this simplified valuation system are shown in *Figure 12B*. According to Equation 12.6, the estimate of the value of state $s$ converges in equilibrium to $v = \overline{w}\,\overline{s}$. Following reward delivery, parameters $w_j$ are modified according to Equation 12.7, where $v$ is taken as the estimated value at the end of simulation of the planning phase on this trial.

In order to simulate the actor, its description has been converted to differential equations in analogous way as in *Figure 6C*. At the end of the planning phase, Gaussian noise with standard deviation $\sigma_a = 2$ was added to all entries of the action vector (to allow exploration), and the action with the highest intensity was 'chosen' by the model. Subsequently, for the chosen action $i$ the intensity was set to $a_i = 1$, while for the other action it was set to $a_{k \neq i} = 0$. For simplicity we did not explicitly simulate the dynamics of the model after the delivery of reward $r$, but we computed the prediction errors in the goal-directed and habit system in an equilibrium (orange Equation 9.6 and Equation 10.1), and updated the parameters. In simulations the learning rate in the valuation system was set to $\alpha_v = 0.5$ on trials with $\delta_v > 0$, and to $\alpha_v = 0.1$ when $\delta_v \leq 0$. Other learning rates were set to: $\alpha_g = 0.1$, $\alpha_h = 0.05$, $\alpha_\Sigma = 0.01$. The remaining parameters of the simulations had the same value as in the previous section.

## Acknowledgements

This work has been supported by MRC grants MC_UU_12024/5, MC_UU_00003/1 and BBSRC grant BB/S006338/1. The author thanks Moritz Moeller and Sashank Pisupati for comments on an earlier version of the manuscript, and Karl Friston, Yonatan Loewenstein, Mark Howe, Friedemann Zenke, Kevin Miller and Peter Dayan for discussion.

## Additional information

### Funding

| Funder | Grant reference number | Author |
|---|---|---|
| Medical Research Council | MC_UU_12024/5 | Rafal Bogacz |
| Medical Research Council | MC_UU_00003/1 | Rafal Bogacz |
| Biotechnology and Biological Sciences Research Council | BB/S006338/1 | Rafal Bogacz |

The funders had no role in study design, data collection and interpretation, or the decision to submit the work for publication.

### Author contributions

Rafal Bogacz, Conceptualization, Formal analysis, Investigation

### Author ORCIDs

Rafal Bogacz  https://orcid.org/0000-0002-8994-1661

### Decision letter and Author response

Decision letter https://doi.org/10.7554/eLife.53262.sa1

Author response https://doi.org/10.7554/eLife.53262.sa2

## Additional files

### Supplementary files
• Transparent reporting form

### Data availability
Matlab codes for all simulations described in the paper are available at MRC Brain Network Dynamics Unit Data Sharing Platform (https://data.mrc.ox.ac.uk/data-set/simulations-action-inference). Users must first create a free account (https://data.mrc.ox.ac.uk/user/register) before they can download the datasets from the site.

The following dataset was generated:

| Author(s) | Year | Dataset title | Dataset URL | Database and Identifier |
|---|---|---|---|---|
| Bogacz R | 2020 | Simulations of action inference | https://data.mrc.ox.ac.uk/data-set/simulations-action-inference | MRC Brain Network Dynamics Unit Data Sharing Platform, simulations-action-inference |

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
