## [Decision Letter]

**Acceptance summary:**

This manuscript presents a novel computational framework of dopamine function: DopAct. It integrates reinforcement learning with ideas from active inference, Bayesian inference, and the recent proposal that goal-directed and habitual systems use different prediction errors for learning. This model is exciting because it accounts for the role of dopamine in reward learning and action planning.

**Decision letter after peer review:**

Thank you for sending your article entitled "Dopamine role in learning and action inference" for peer review at *eLife*. Your article is being evaluated by two peer reviewers, and the evaluation is being overseen by a Reviewing Editor and Kate Wassum as the Senior Editor.

Both reviewers and the editorial team felt that your manuscript describing a new model for dopamine in action planning and habit learning is very interesting.

However, both reviewers raised several major concerns. The points are detailed in the individual critiques (see below). We would appreciate your response to each point raised by both reviewers, paying special attention to the following most essential points.

1) Explain and demonstrate whether and how the model can account for the well-documented shift in dopamine bursts from reward to predictive stimulus with learning (reviewer #2, point 2).

2) Explain and demonstrate whether and how the model can account for extinction (reviewer #1, point 5-6; reviewer #2, point 2). This point is related to the general question of how the goal-directed and habit system interact, and how the goal-directed system can regain control over action selection.

3) The paper itself will require several major changes. It will be important to describe the model more clearly (including the valuation system) so that it can be fully understood (reviewer #1, points 1-3; reviewer #2, point 1 and 3), and to better connect the model to the neuroscience literature (reviewer #2, point 2).

Reviewer #1:

This paper presents a theoretical framework, referred to as DopAct, to account for the role of dopamine (DA) in learning and action planning. While this framework makes a set of novel assumptions, it draws heavily on classical reinforcement learning (e.g. Sutton and Barto, 1998), Friston's (2010) active inference theory, Miller et al.'s (2019) theory of habit formation and Solway and Botvinick's (2012) Bayesian inference theory of action selection. Following current thinking, DopAct proposes that different cortico-striatal loops targeted by distinct DAergic neuronal populations underpin outcome valuation, goal-directed action selection, and habit formation functions. Critically, however, DopAct extends the role of DA beyond that of driving learning and into action planning, although this is restricted to the goal-directed, but not the habit system. The major contribution of the paper is thus to provide a formal integration of (variants of) multiple preexisting theories and to propose a particular neural instantiation in the basal ganglia. My concerns are as follows.

1) I found the paper somewhat difficult to follow due to the fact that critical pieces of information necessary to understand how the various components of DopAct operate and interact with each other appear fragmented and scattered across the paper. In addition, the writing style in the paper is at times unpolished (by way of example, consider the sentence starting with "The model has been simulated…”…, which undermines clarity.

2) In my opinion, the paper is made unnecessarily complex by the amalgamation of multiple theoretically independent ideas, not all of which are equally essential to the central message or formally implemented. For instance, at the beginning of the Results, the reader is given an overview of the DopAct framework, starting with the valuation system. Therein, it is stated that DopAct departs from the classical reinforcement learning principle of reward maximization, replacing it instead with the proposal that the valuation system computes the amount of resources (e.g., food) necessary to bring the animal to its optimal reserve levels. This notion is not, however, formally developed in the paper and is in practice relegated to a future direction. Indeed, all simulations assume low levels of reserves and thus are undistinguishable from standard reinforcement learning. For this reason, I think introducing this topic at the outset (or at all) only acts as a distractor of the main points of the DopAct framework and thus I would encourage the author to reserve this important yet here ancillary distinction for a future paper. When doing so, evidence should be provided to justify this assumption (e.g. evidence that a moderately hungry animal would choose a small over a large reward if the former is sufficient to restore desired reserve levels).

3) Insufficient details are given as to how prediction errors operate within the valuation system and how these errors relate to those being computed in the goal-directed system. Are expectations in the valuation system only generated by actions planned by the goal-directed system, or can states generate their own expectations without the mediation of actions? If the latter, how do the two types of expectations (cue-based vs. action-based) interact, if at all, during learning to determine the computation of prediction errors in each system? This is particularly important early on in training, while it is unclear to the agent whether the reward is brought about by an action or predicted by a cue and delivered independently of any action. In connection with these questions, the paper announces early on (Introduction section), that Friston's (2010) notions of active inference will be followed, whereby prediction errors can be minimized either by learning to update expectations (no intervening action) or by learning act so as to satisfy those expectations. This point would seem to be particularly relevant to Pavlovian-instrumental interactions during learning, yet such a theme is not developed thereafter in the paper as the focus is placed on action-generated reward expectations (which once again seemed somewhat of a red herring for the reader).

4) If I understood correctly, the goal-directed system is in charge of simultaneously computing reward expectation (generated from an action plan) as well as the required action intensity (at least before the habit system takes over). It would seem, however, that these two variables might not always correlate. For example, two equally large rewards might generate equally large expectations, but one of them might require an action of weaker intensity than the other. How would DopAct deal with such scenarios if action intensity and reward expectation are one and the same thing from the viewpoint of the goal-directed system?

5) Since Thorndike, traditional views of habit learning have proposed the formation of an S-R association stamped on by reinforcement, but excluding the reward itself from the associative representations involved. This view allows habitual animals to become insensitive to outcome devaluation while remaining sensitive to changes in the probability of reward given the action (e.g. extinction). Following Miller et al., 2019, this paper departs from this view by assuming that habit learning is driven by a different, outcome-independent, purely action-based prediction error, calculated as the difference between the habitual (and therefore expected) action and the action actually executed. Early on in training, the action executed is determined by the goal-directed system, but over the course of training control over action execution is yielded to the faster, more computationally efficient habit system. Given this departure from the conventional theoretical framework, a discussion of how these two distinct theoretical assumptions on habit compare to each other would be desirable.

6) Still on the topic of habit formation, I found the paper lacking in clarity as to how the relative contribution of the goal-directed and habit systems to action selection is negotiated when response-outcome contingencies change after habitual responding has been strongly established. First, what exactly happens to the goal-directed system once the habit system takes over? Does it become dormant in the model, or does it do its regular computations (which would seem inefficient)? And second, if the weight of this relative contribution shifts over training so as to favor habit-based responding, how can an animal ever get out of habitual responding unless control is yielded back to the goal-directed system? Based on the simulations show in Figure 7, it would seem that the DopAct has a hard time ever exiting habit mode even after 500 trials of reward omission. Note, however, that evidence (e.g. Balleine, Killcross and Dickinson, 2003) indicates that lesions that cause animals to become habitual (as measured by insensitivity to outcome devaluation) do not necessarily disrupt extinction learning, which would support traditional S-R views of habit. Later in the paper, however, while discussing reversal learning, it is explained that with sufficient reversal training the goal-directed system prevails and manages to retrain the habit system to choose the appropriate response. How does this happen if the action executed (which is what feeds into the prediction error of the habit system) is now determined by the habit system itself? Is it the case that the goal-directed system retains greater control over responding even after strong habits have been established? Such an assumption would indeed allow such retraining of the habit system by the goal-directed system, but it would seem to come at the price of the agent never entirely losing goal-directed behavior (i.e., never quite become habitual).

Reviewer #2:

Summary: This paper presents a new model of goal-directed and habitual learning based on active inference. The paper provides an interesting, integrative model that builds on several different strands of prior work. In addition, the model's approach to habitual and goal-directed learning provide some ideas of how the dopamine system might be parcellated computationally. Moreover, the model tackles a thorny (and largely unaddressed computationally) question of how dopamine can be simultaneously involved in reward prediction and model planning. As it stands, however, I don't think this paper does enough to both explain how the model works, nor justify its assumptions and empirical suitability. I unpack these points below.

1) The model has many assumptions and parameters, which seem not sufficiently motivated and constrained by the target behaviour and underlying neuroscience. For example, the valuation system is hardly explained and relies on resource levels that are not used in any of the simulations. Similarly, pointers are made to the neuroscience, but mostly in a speculative way, rather than a firmer linking between computational elements and underlying anatomy/physiology. What could possibly show that this model is wrong? That's not adequately clear. Relatedly, too much is left to future work-perhaps compounding the impression that the model is not sufficiently justified/supported as yet.

2) The range of applications used to justify the model is very narrow-only two basic behavioural cases are undertaken, while some of the fundamental results in this area (e.g., the shift in dopamine burst from Schultz, Dayan, Montague, 1997 in Science) are not obviously replicated. The claim is noted that the analogy to the TD model means that similar learning should occur, but that is not really shown (and very much not obvious from the exposition). In addition, the major point about how action intensity becomes habitized misses a key distinction between variable-interval and variable-ration schedules (VI schedules with lower response rates produce faster habits)-how would this model possibly account for this (it would seem to go counter the mechanism)? Even the simulated result (on omission) seems to overstate the case in that even after 500 trials, there is no extinction (Figure 7A).

3) The model seems to assume computationally intensive (and complex) operations for the organism to solve even simple problems. For example, the model requires sophisticated action planning to terminate the dopamine burst even on the very first trial, which would seem to require further justification. This does suggest a testable prediction, though: does the dopamine burst get shorter as the planning gets more efficient? When the only learning requires an intensity adjustment for a response (as modelled), this planning mechanism seems potentially plausible, but not clear how that would generalize more broadly. As a result, even the algorithmic level in the paper reads very much as a computational account (what the system should achieve, rather than how it does). There would seem to be some insights there in how food delivery is transformed into a perceived reward, but that aspect is not drawn out.

[Editors' note: further revisions were suggested prior to acceptance, as described below.]

Thank you for resubmitting your article "Dopamine role in learning and action inference" for consideration by *eLife*. Your revised article has been reviewed by two peer reviewers, and the evaluation has been overseen by a Reviewing Editor and Kate Wassum as the Senior Editor. The following individuals involved in review of your submission have agreed to reveal their identity: Philipp Schwartenbeck (Reviewer #4).

The reviewers have discussed the reviews with one another and the Reviewing Editor has drafted this decision to help you prepare a revised submission.

Summary:

All reviewers agreed that the revised manuscript is much improved. However, reviewers also felt that the overall structure of the manuscript could still be improved in terms of simplicity and clarity, and that additional clarifications were needed regarding a few select issues.

Revisions:

1) Reviewers agreed that the paper is still too dense and it would be important to clarify the writing to make it more concise. Reviewers suggested highlighting the general aspects of the model and simplifying specifics of the implementation. They suggested moving some of the details to an appendix, as well as providing concise summaries, and stating what are the key general points and what are more detailed implementational aspects for a specific simulation. Further, the discussion of the novel theoretical and empirical predictions is very rich at the moment but could be summarized and simplified to highlight some of the main points. This could include, for instance, the teaching signals in the valuation (RPE independent of action plan?), goal-directed (RPE dependent on action plan?) and habit (action prediction errors?) system, and the need for the goal-directed system to learn about reversals or the resulting effects on behavior (including blocking of the different systems).

2) A specific issue that needs to be clarified is how an animal can exit habit mode. Specifically, the author states: "On later trials the action is jointly determined by the habit and goal-directed system (Figure 1B), and their relative contributions depend on their levels of confidence." It is still unclear how exactly a habitized animal would exit habit mode if the relative contributions of the goal-directed and habit systems are determined by their level of confidence. How is confidence formally estimated? For instance, if the expectancies of a habitized animal are violated, this assumption would seem to predict that the goal-directed system should lose some confidence. If so, it would seem to follow that the goal-directed system should yield even more control to the misguided but confident habit system. In DopAct an animal can get out of a habit only through exploration (Results final paragraph). It is unclear whether and how exploration is promoted by prediction errors, or whether it is treated as a constant in the model. This is critical because the rate of exploration in habit-dominated behaviors would be expected to be low once the learning agent has settled on a stable solution. Thus, given a change in contingencies, a confidence-based arbitration between the goal-directed and habit systems would seem to further privilege the habit system while the goal-directed system waits for an opportunity to rewire itself into a more adaptive set of associations. At what point will those new associations carry more confidence than those deeply entrenched in the habit system?

3) A key prediction relates to the dissociation between the goal-directed and habitual system learning from reward prediction errors and action prediction errors, respectively. Their roles and dynamics could be described more clearly and currently these descriptions are quite scattered around the text. Does this model predict an independent signature of reward and action prediction errors in a task where those two teaching signals can be fully dissociated (e.g., rewards without actions or the other way around)? Further, is the arbitration between those two modes based on their relative uncertainty? Is it just the decrease in uncertainty that drives actions to become habitual over time, or is there also some inherent complexity penalty for the goal-directed system, as for example described in the active inference framework (e.g. FitzGerald, Dolan and Friston, 2014)?

4) Another point that requires some clarification is "Once an action plan has been formulated, the animal starts to expect the available reward, and the dopamine level encoding the prediction error decreases." Despite the substantial expansion, this section on active inference is still confusing. This may in part be due to semantics. For example, is by "an action plan has been formulated" actually meant "an action plan has been implemented"? It is hard to see how the mere formulation (i.e., elaboration) of an action plan could affect the kind of "change in the world" that is being proposed to contribute to reducing prediction errors (alongside learning). A related question: would habitual responses also reduce prediction errors, given that by definition they do not generate reward expectancies?

5) There is an interesting distinction between the valuation and the goal-directed system, since both display reward prediction errors but the goal-directed reward prediction errors are contingent on action plans. Would that predict an absence of reward prediction errors in the goal directed system if the agent cannot perform actions, or is forced to take actions that are likely to result in no rewards?

6) Related to this point the author states: "in the DopAct framework the expectation of reward only arises from formulating a plan to achieve it." This statement is odd in light that the valuation system, which is part of the DopAct framework, is proposed to compute reward expectancies on the bases of antecedent stimuli without the mediation of action plans. Should this sentence read instead "in the goal-directed system the expectation of reward only arises from formulating a plan to achieve it."? This would be consistent with the idea that, while action plans may be a fundamental component of generating reward expectations, animals can also acquire such expectancies through stimulus-stimulus (or state-state) learning. It would also allow DopAct to account for recent evidence in the sensory preconditioning paradigm indicating that associations between neutral stimuli are promoted by DA stimulation without the mediation of any action plan. These findings could not be accommodated by DopAct if the sentence “…but in the DopAct framework the expectation of reward only arises from formulating a plan to achieve it.” was true.

7) Given that the model is fully Bayesian, agents not only have access to their beliefs about actions and rewards but also their uncertainty in these beliefs. Does the model make any interesting predictions for signals that reflect the reduction of uncertainty about actions and rewards? Is there also an effect for the reduction of state uncertainty? The latter point would be particularly interesting with respect to recent reports about dopaminergic signals for state identity that are orthogonal to reward (e.g. Takahashi et al., 2017). If not, which changes to the model architecture would be necessary to account for such signals?

---

## [Author Response]

However, both reviewers raised several major concerns. The points are detailed in the individual critiques (see below). We would appreciate your response to each point raised by both reviewers, paying special attention to the following most essential points.1) Explain and demonstrate whether and how the model can account for the well-documented shift in dopamine bursts from reward to predictive stimulus with learning (reviewer #2, point 2).

A new Figure 7B has been added demonstrating that the model can reproduce the shift in dopamine burst from reward to predictive stimulus.

2) Explain and demonstrate whether and how the model can account for extinction (reviewer #1, point 5-6; reviewer #2, point 2). This point is related to the general question of how the goal-directed and habit system interact, and how the goal-directed system can regain control over action selection.

A new Figure 8 has been added, which demonstrates that the model can produce extinction, and qualitatively reproduce key features of extinction behaviour seen in two experimental studies. It has also been clarified how the goal-directed and habit systems interact: Figure 1B and text describing it (paragraph four “Overview of the framework”) have been modified to clarify that even when an action becomes habitual, both the goal-directed and habit systems contribute to the action planning, but their contribution are weighted by their confidence. A paragraph (final paragraph in the Results) has also been added discussing the mechanisms underlying reversal in the model.

3) The paper itself will require several major changes. It will be important to describe the model more clearly (including the valuation system) so that it can be fully understood (reviewer #1, points 1-3; reviewer #2, point 1 and 3), and to better connect the model to the neuroscience literature (reviewer #2, point 2).

I have been convinced by comment 2 of reviewer 1 that the conceptual description of validation system at the start of the Results section distracts from key points of the manuscript, and I followed his/her recommendation to shorten this description, so it is now contained within just a single paragraph. At the same time, details of the presented simulations of the valuation system have been added (Figure 12A and text in Materials and methods). Furthermore, additional simulations of classical paradigms suggested by the reviewers that have been added (classical conditioning – Figure 7B, devaluation – Figure 8A, and Pavlovian-instrumental transfer – Figure 8B) to better connect the manuscript with the neuroscience literature.

Reviewer #1:This paper presents a theoretical framework, referred to as DopAct, to account for the role of dopamine (DA) in learning and action planning. While this framework makes a set of novel assumptions, it draws heavily on classical reinforcement learning (e.g. Sutton and Barto, 1998), Friston's (2010) active inference theory, Miller et al.'s (2019) theory of habit formation and Solway and Botvinick's (2012) Bayesian inference theory of action selection. Following current thinking, DopAct proposes that different cortico-striatal loops targeted by distinct DAergic neuronal populations underpin outcome valuation, goal-directed action selection, and habit formation functions. Critically, however, DopAct extends the role of DA beyond that of driving learning and into action planning, although this is restricted to the goal-directed, but not the habit system. The major contribution of the paper is thus to provide a formal integration of (variants of) multiple preexisting theories and to propose a particular neural instantiation in the basal ganglia. My concerns are as follows.1) I found the paper somewhat difficult to follow due to the fact that critical pieces of information necessary to understand how the various components of DopAct operate and interact with each other appear fragmented and scattered across the paper.

The descriptions of the models (that in the previous version was spread between Results and Materials and method sections) has been integrated and is presented in the Results. The figures with details of the models have also been moved to the Results, and integrated with the figures in Results (new Figures 6, 9, 10).

In addition, the writing style in the paper is at times unpolished (by way of example, consider the sentence starting with "The model has been simulated…”, which undermines clarity.

The sentence mentioned above is no longer included in the manuscript, because the simulations of the omission protocol it described have been replaced by the simulations aiming to replicate experimental data from extinction paradigms. The manuscript has been carefully re-read and re-checked.

2) In my opinion, the paper is made unnecessarily complex by the amalgamation of multiple theoretically independent ideas, not all of which are equally essential to the central message or formally implemented. For instance, at the beginning of the Results, the reader is given an overview of the DopAct framework, starting with the valuation system. Therein, it is stated that DopAct departs from the classical reinforcement learning principle of reward maximization, replacing it instead with the proposal that the valuation system computes the amount of resources (e.g., food) necessary to bring the animal to its optimal reserve levels. This notion is not, however, formally developed in the paper and is in practice relegated to a future direction. Indeed, all simulations assume low levels of reserves and thus are undistinguishable from standard reinforcement learning. For this reason, I think introducing this topic at the outset (or at all) only acts as a distractor of the main points of the DopAct framework and thus I would encourage the author to reserve this important yet here ancillary distinction for a future paper. When doing so, evidence should be provided to justify this assumption (e.g. evidence that a moderately hungry animal would choose a small over a large reward if the former is sufficient to restore desired reserve levels).

I would like to thank the reviewer for this great observation and suggestion. In the revised version of the manuscript, the description of the valuation system has been shortened (to avoid making unnecessary assumptions) and is now contained within a single paragraph (paragraph two “Overview of the framework”).

3) Insufficient details are given as to how prediction errors operate within the valuation system and how these errors relate to those being computed in the goal-directed system. Are expectations in the valuation system only generated by actions planned by the goal-directed system, or can states generate their own expectations without the mediation of actions? If the latter, how do the two types of expectations (cue-based vs. action-based) interact, if at all, during learning to determine the computation of prediction errors in each system? This is particularly important early on in training, while it is unclear to the agent whether the reward is brought about by an action or predicted by a cue and delivered independently of any action. In connection with these questions, the paper announces early on (Introduction section), that Friston's (2010) notions of active inference will be followed, whereby prediction errors can be minimized either by learning to update expectations (no intervening action) or by learning act so as to satisfy those expectations. This point would seem to be particularly relevant to Pavlovian-instrumental interactions during learning, yet such a theme is not developed thereafter in the paper as the focus is placed on action-generated reward expectations (which once again seemed somewhat of a red herring for the reader).

In the revised version of the manuscript, more detailed simulations of the valuation system were added, which demonstrate in detail how the prediction errors are generated. A new Figure 12A and text describing it have been added showing how the temporal difference learning algorithm has been implemented in the model.

Furthermore, a simulation of the model in Pavlovian-instrumental transfer paradigm has been added which demonstrates that the model can replicate experimentally observed pattern of behaviour, as shown in Figure 8B, and explained in Results and Materials and methods. Is has been clarified there that during operant conditioning reward expectation computed by the valuation system drives action planning, while during classical conditioning it may provide expectations without triggering actions. It has been also clarified how the interactions between the two types of expectations allow the model to reproduce Pavlovian instrumental transfer.

It has been clarified how prediction errors in the goal-directed system are reduced by both planning and learning: Figure 2 has been extended to include 2 panels that highlight these two ways of minimizing prediction errors, and the description has been added in the text (final paragraph “Overview of the framework”).

4) If I understood correctly, the goal-directed system is in charge of simultaneously computing reward expectation (generated from an action plan) as well as the required action intensity (at least before the habit system takes over). It would seem, however, that these two variables might not always correlate. For example, two equally large rewards might generate equally large expectations, but one of them might require an action of weaker intensity than the other. How would DopAct deal with such scenarios if action intensity and reward expectation are one and the same thing from the viewpoint of the goal-directed system?

A footnote has been added clarifying that in the above case the action with lower required intensity will have a higher value of parameter q. In this way, both actions will produce the same expectation, which is proportional to qa rather than q itself, but the two rewards will result in different action intensities, which in the model depend on both v and q.

5) Since Thorndike, traditional views of habit learning have proposed the formation of an S-R association stamped on by reinforcement, but excluding the reward itself from the associative representations involved. This view allows habitual animals to become insensitive to outcome devaluation while remaining sensitive to changes in the probability of reward given the action (e.g. extinction). Following Miller et al., 2019, this paper departs from this view by assuming that habit learning is driven by a different, outcome-independent, purely action-based prediction error, calculated as the difference between the habitual (and therefore expected) action and the action actually executed. Early on in training, the action executed is determined by the goal-directed system, but over the course of training control over action execution is yielded to the faster, more computationally efficient habit system. Given this departure from the conventional theoretical framework, a discussion of how these two distinct theoretical assumptions on habit compare to each other would be desirable.

The paragraph discussing the relationship of DopAct framework to traditional computational models of habit formation has been substantially extended. Furthermore, a new paragraphs has been added that compares the accounts of habit formation in DopAct and traditional reinforcement learning models with experimental data (Discussion section).

6) Still on the topic of habit formation, I found the paper lacking in clarity as to how the relative contribution of the goal-directed and habit systems to action selection is negotiated when response-outcome contingencies change after habitual responding has been strongly established. First, what exactly happens to the goal-directed system once the habit system takes over? Does it become dormant in the model, or does it do its regular computations (which would seem inefficient)? And second, if the weight of this relative contribution shifts over training so as to favor habit-based responding, how can an animal ever get out of habitual responding unless control is yielded back to the goal-directed system? Based on the simulations show in Figure 7, it would seem that the DopAct has a hard time ever exiting habit mode even after 500 trials of reward omission. Note, however, that evidence (e.g. Balleine, Killcross and Dickinson, 2003) indicates that lesions that cause animals to become habitual (as measured by insensitivity to outcome devaluation) do not necessarily disrupt extinction learning, which would support traditional S-R views of habit. Later in the paper, however, while discussing reversal learning, it is explained that with sufficient reversal training the goal-directed system prevails and manages to retrain the habit system to choose the appropriate response. How does this happen if the action executed (which is what feeds into the prediction error of the habit system) is now determined by the habit system itself? Is it the case that the goal-directed system retains greater control over responding even after strong habits have been established? Such an assumption would indeed allow such retraining of the habit system by the goal-directed system, but it would seem to come at the price of the agent never entirely losing goal-directed behavior (i.e., never quite become habitual).

New simulations in Figure 8 have been added demonstrating that the model can produce extinction. The occurrence of extinction depends on the values of parameters of the simulated model, and in the previous version of the manuscript the parameters were chosen to replicate the simulations shown in Figure 4 in the paper by Miller et al., 2019, where their model also did not show extinction after extensive training (left panel in Figure 4 in their paper). However, thanks to reviewers’ comments I realized that such behaviour is not observed experimentally, hence in the revised version, different parameters were used for which the model reproduces profiles of extinction observed in behavioural experiments (re-plotted in top displays on Figure 8).

Furthermore, Figure 1B and text have been modified to clarify that even if an action becomes habitual, both goal-directed and habit systems contribute to action selection, and their contribution depends on their relative confidence. A paragraph (penultimate paragraph Results section) has also been added discussing the mechanism underlying reversal in the model, which clarifies that some contribution of the goal-directed system to action planning is necessary for reversal to take place, as pointed out by the reviewer above.

A sentence has been added in a section discussing future work mentioning the role of amygdala in habit formation and citing the paper by Balleine et al., 2003, highlighted by the reviewer.

Reviewer #2:[…] 1) The model has many assumptions and parameters, which seem not sufficiently motivated and constrained by the target behaviour and underlying neuroscience. For example, the valuation system is hardly explained and relies on resource levels that are not used in any of the simulations.

Following a suggestion of reviewer 2, the description of the valuation system has been shortened (to avoid making unnecessary assumptions) and is now contained within a single paragraph of the Results. On the other hand, more details were provided on the operation of the valuation system in the simulations (new Figure 12A, and text).

Similarly, pointers are made to the neuroscience, but mostly in a speculative way, rather than a firmer linking between computational elements and underlying anatomy/physiology. What could possibly show that this model is wrong? That's not adequately clear. Relatedly, too much is left to future work-perhaps compounding the impression that the model is not sufficiently justified/supported as yet.

Additional simulations of classical paradigms (conditioning, extinction, Pavlovian-instrumental transfer) have been added to better connect the manuscript with neuroscience literature and are presented in new Figures 7B, 8A and 8B. Furthermore, the section discussing experimental predictions has been reorganized such that it starts with discussing the two most critical predictions that could allow verification or falsification of the model. Specific experiments have been described that could be performed to test these predictions.

2) The range of applications used to justify the model is very narrow-only two basic behavioural cases are undertaken, while some of the fundamental results in this area (e.g., the shift in dopamine burst from Schultz, Dayan, Montague, 1997 in Science) are not obviously replicated. The claim is noted that the analogy to the TD model means that similar learning should occur, but that is not really shown (and very much not obvious from the exposition).

The simulations of the shift of dopaminergic response from reward to conditioned stimulus have been added in the new Figure 7B and text.

In addition, the major point about how action intensity becomes habitized misses a key distinction between variable-interval and variable-ration schedules (VI schedules with lower response rates produce faster habits)-how would this model possibly account for this (it would seem to go counter the mechanism)?

This phenomenon has been discussed in detail in the paper by Miller et al., 2019, which shows how their model reproduces this effect in simulations (see Figure 5 in paper by Miller et al., 2019). Due to a conceptual similarity of the presented model to the model by Miller et al., these two models account for the phenomenon in related ways. A paragraph has been added in Discussion summarizing this account.

Even the simulated result (on omission) seems to overstate the case in that even after 500 trials, there is no extinction (Figure 7A).

New simulations in Figure 8 have been added demonstrating that the model can produce extinction. The occurrence of extinction depends on the values of parameters of the simulated model, and in the previous version of the manuscript the parameters were chosen to replicate the simulations shown in Figure 4 in the paper by Miller et al., 2019, where their model also did not show extinction after extensive training (left panel in Figure 4 in their paper). However, thanks to reviewers’ comments I realized that such behaviour is not observed experimentally, hence in the revised version, different parameters were used for which the model reproduces profiles of extinction observed in behavioural experiments (re-plotted in top displays on Figure 8).

3) The model seems to assume computationally intensive (and complex) operations for the organism to solve even simple problems. For example, the model requires sophisticated action planning to terminate the dopamine burst even on the very first trial, which would seem to require further justification. This does suggest a testable prediction, though: does the dopamine burst get shorter as the planning gets more efficient? When the only learning requires an intensity adjustment for a response (as modelled), this planning mechanism seems potentially plausible, but not clear how that would generalize more broadly. As a result, even the algorithmic level in the paper reads very much as a computational account (what the system should achieve, rather than how it does).

I would like to thank the reviewer for noticing this interesting prediction, it has been added to the section listing experimental predictions in the revised version of the manuscript.

There would seem to be some insights there in how food delivery is transformed into a perceived reward, but that aspect is not drawn out.

Such insight could be provided by a detailed model of the valuation system, but following the recommendation of reviewer 2, the description of the valuation system will be left for a future study.

[Editors' note: further revisions were suggested prior to acceptance, as described below.]

Revisions:1) Reviewers agreed that the paper is still too dense and it would be important to clarify the writing to make it more concise. Reviewers suggested highlighting the general aspects of the model and simplifying specifics of the implementation. They suggested moving some of the details to an appendix, as well as providing concise summaries, and stating what are the key general points and what are more detailed implementational aspects for a specific simulation.

I thank the reviewers for these great suggestions. The manuscript has been substantially reorganized to focus on general aspects of the model and to reduce or remove from Results the details of implementation:

The Abstract has been simplified to better emphasize the key points of the paper.To make the paper easier to navigate, it has been divided into a larger number of shorter sections, with more informative titles.Summary of key points has been added at the starts of multiple paragraphs.A new paragraph has been added summarizing the key feature of the algorithm in DopAct (subsection “Algorithm for planning and learning” final paragraph).The details of a possible way the prediction errors in the goal-directed system could be computed have been deleted are replaced by the key point that such computation would be local in the basal ganglia network. This also allowed simplification of network diagrams in Figures 5, 6 and 10.Description of Figure 6 has been shortened, and some of the details have been moved from the main text to the figure caption.Description of Figure 7C has been shortened and organized around the key messages rather than individual displays.Description of details of devaluation simulations has been moved from Results to Materials and methods.Description of hyperpriors of variance parameters have been moved from Results to Materials and methods, and shortened.A paragraph discussing the details of units of various terms in Equation 9.9 has been deleted.An uninteresting simulation of the valuation system has been removed from Figure 11C, which allowed shortening Materials and methods, and simplifying Figure 12B.

Further, the discussion of the novel theoretical and empirical predictions is very rich at the moment but could be summarized and simplified to highlight some of the main points. This could include, for instance, the teaching signals in the valuation (RPE independent of action plan?), goal-directed (RPE dependent on action plan?) and habit (action prediction errors?) system, and the need for the goal-directed system to learn about reversals or the resulting effects on behavior (including blocking of the different systems).

The Discussion section has been simplified and reorganized:

The description of the relationship of the DopAct to other theories has been shorten and focused on key points.Section “Relationship to experimental data” has been substantially shortened, focussed on key experimental data, and reorganized around predictions rather than discussed studies.Different paragraphs from the previous version of Discussion concerned with habit formation have been gathered in a new section “Mechanisms of habitual behaviour”.Section “Direction of future work” has been shortened.

2) A specific issue that needs to be clarified is how an animal can exit habit mode. Specifically, the author states: "On later trials the action is jointly determined by the habit and goal-directed system (Figure 1B), and their relative contributions depend on their levels of confidence." It is still unclear how exactly a habitized animal would exit habit mode if the relative contributions of the goal-directed and habit systems are determined by their level of confidence. How is confidence formally estimated?

Throughout the manuscript word “confidence” has been replaced by “certainty”, as the concept of uncertainty is formally defined in the paper.

For instance, if the expectancies of a habitized animal are violated, this assumption would seem to predict that the goal-directed system should lose some confidence. If so, it would seem to follow that the goal-directed system should yield even more control to the misguided but confident habit system.

A new paragraph has been added discussing this effect and pointing that it is only transient (Results final paragraph).

In DopAct an animal can get out of a habit only through exploration (Results final paragraph). It is unclear whether and how exploration is promoted by prediction errors, or whether it is treated as a constant in the model. This is critical because the rate of exploration in habit-dominated behaviors would be expected to be low once the learning agent has settled on a stable solution.

The mechanisms allowing a reversal in the model have been explained in more detail, and it has been emphasized that the amount of noise resulting in exploration was constant in simulations.

Thus, given a change in contingencies, a confidence-based arbitration between the goal-directed and habit systems would seem to further privilege the habit system while the goal-directed system waits for an opportunity to rewire itself into a more adaptive set of associations. At what point will those new associations carry more confidence than those deeply entrenched in the habit system?

A new paragraph has been added to section “Mechanisms of habitual behaviour” discussing how the goal-directed system can regain control from the habit system.

3) A key prediction relates to the dissociation between the goal-directed and habitual system learning from reward prediction errors and action prediction errors, respectively. Their roles and dynamics could be described more clearly and currently these descriptions are quite scattered around the text.

A paragraph has been added in section “Overview of the framework” which highlights that the two systems learn on the basis of different prediction errors. Also, Figure 1D has been modified to illustrate the process of learning in the habit system.

Does this model predict an independent signature of reward and action prediction errors in a task where those two teaching signals can be fully dissociated (e.g., rewards without actions or the other way around)?

This is indeed the key prediction of the model, and such pattern of activity was indeed observed by Howe and Dombeck. To increase prominence of the discussion of this key study, it has been moved to the second paragraph of section “Relationship to experimental data”.

Further, is the arbitration between those two modes based on their relative uncertainty? Is it just the decrease in uncertainty that drives actions to become habitual over time, or is there also some inherent complexity penalty for the goal-directed system, as for example described in the active inference framework (e.g. FitzGerald, Dolan and Friston, 2014)?

A sentence summarizing the key idea from the above paper has been added to section “Mechanisms of habitual behaviour”.

4) Another point that requires some clarification is "Once an action plan has been formulated, the animal starts to expect the available reward, and the dopamine level encoding the prediction error decreases." Despite the substantial expansion, this section on active inference is still confusing. This may in part be due to semantics. For example, is by "an action plan has been formulated" actually meant "an action plan has been implemented"? It is hard to see how the mere formulation (i.e., elaboration) of an action plan could affect the kind of "change in the world" that is being proposed to contribute to reducing prediction errors (alongside learning).

Thank you for these great questions. A paragraph discussing them has been added in section “Overview of the framework”.

A related question: would habitual responses also reduce prediction errors, given that by definition they do not generate reward expectancies?

Yes, this effect is visible in the right display of Figure 7C, and a sentence has been added discussing it.

5) There is an interesting distinction between the valuation and the goal-directed system, since both display reward prediction errors but the goal-directed reward prediction errors are contingent on action plans. Would that predict an absence of reward prediction errors in the goal directed system if the agent cannot perform actions, or is forced to take actions that are likely to result in no rewards?

The DopAct framework predicts that such manipulation will result in prolong prediction errors, and this has been added in section “Experimental predictions”.

6) Related to this point the author states: "in the DopAct framework the expectation of reward only arises from formulating a plan to achieve it." This statement is odd in light that the valuation system, which is part of the DopAct framework, is proposed to compute reward expectancies on the bases of antecedent stimuli without the mediation of action plans. Should this sentence read instead "in the goal-directed system the expectation of reward only arises from formulating a plan to achieve it."? This would be consistent with the idea that, while action plans may be a fundamental component of generating reward expectations, animals can also acquire such expectancies through stimulus-stimulus (or state-state) learning. It would also allow DopAct to account for recent evidence in the sensory preconditioning paradigm indicating that associations between neutral stimuli are promoted by DA stimulation without the mediation of any action plan. These findings could not be accommodated by DopAct if the sentence “…but in the DopAct framework the expectation of reward only arises from formulating a plan to achieve it.” was true.

Phrase “in the goal-directed system” has been added to the sentence quoted above.

7) Given that the model is fully Bayesian, agents not only have access to their beliefs about actions and rewards but also their uncertainty in these beliefs. Does the model make any interesting predictions for signals that reflect the reduction of uncertainty about actions and rewards?

A paragraph describing such prediction has been added to section “Relationship to experimental data”.

Is there also an effect for the reduction of state uncertainty? The latter point would be particularly interesting with respect to recent reports about dopaminergic signals for state identity that are orthogonal to reward (e.g. Takahashi et., 2017). If not, which changes to the model architecture would be necessary to account for such signals?

A summary of this study and the description of an extension of the model that could capture these observations has been added to section “Directions for future work”.